# Identification of a novel mechanism of blood–brain communication during peripheral inflammation via choroid plexus-derived extracellular vesicles

Sriram Balusu[1,2,3,4], Elien Van Wonterghem[1,2], Riet De Rycke[1,2], Koen Raemdonck[5], Stephan Stremersch[5], Kris Gevaert[3,4], Marjana Brkic[1,2,6], Delphine Demeestere[1,2], Valerie Vanhooren[1,2], An Hendrix[7], Claude Libert[1,2,†] & Roosmarijn E Vandenbroucke[1,2,*,†]

## Abstract

Here, we identified release of extracellular vesicles (EVs) by the choroid plexus epithelium (CPE) as a new mechanism of blood–brain communication. Systemic inflammation induced an increase in EVs and associated pro-inflammatory miRNAs, including miR-146a and miR-155, in the CSF. Interestingly, this was associated with an increase in amount of multivesicular bodies (MVBs) and exosomes per MVB in the CPE cells. Additionally, we could mimic this using LPS-stimulated primary CPE cells and choroid plexus explants. These choroid plexus-derived EVs can enter the brain parenchyma and are taken up by astrocytes and microglia, inducing miRNA target repression and inflammatory gene up-regulation. Interestingly, this could be blocked *in vivo* by intracerebroventricular (icv) injection of an inhibitor of exosome production. Our data show that CPE cells sense and transmit information about the peripheral inflammatory status to the central nervous system (CNS) via the release of EVs into the CSF, which transfer this pro-inflammatory message to recipient brain cells. Additionally, we revealed that blockage of EV secretion decreases brain inflammation, which opens up new avenues to treat systemic inflammatory diseases such as sepsis.

**Keywords** blood–brain barrier; choroid plexus; exosomes; extracellular vesicles; sepsis
**Subject Category** Immunology

## Introduction

The choroid plexus epithelium (CPE) is a unique single layer of epithelial cells situated at the interface of the blood and the cerebrospinal fluid (CSF) and forms the blood-CSF barrier (BCSFB). In recent years, the BCSFB has gained increasing attention, especially its role in inflammatory and neurodegenerative diseases (Vandenbroucke *et al*, 2012; Brkic *et al*, 2015; Demeestere *et al*, 2015a,b; Balusu *et al*, 2016; Gorlé *et al*, 2016). Systemic inflammatory response syndrome (SIRS) is a systemic inflammatory disease caused by, for example, trauma, burns, and infection. When infection is suspected, SIRS is called sepsis. Until now, treatment is mainly limited to antibiotics and support of vital functions, so understanding its pathology is of extreme importance for finding new therapeutic targets (Williams, 2012). We recently showed that systemic inflammatory conditions, such as SIRS and sepsis, compromise BCSFB barrier functionality *in vivo*, allowing leakage from the blood into the central nervous system (CNS) via the CSF (Vandenbroucke *et al*, 2012).

The BCSFB is not only an anatomical barrier but also a dynamic tissue that expresses multiple transporters, receptors and enzymes. Indeed, the CPE plays a vital role in maintaining brain homeostasis by producing CSF, a transparent, colorless fluid composed of water, ions, and proteins. CSF is crucial for preserving the chemical microenvironment of the CNS and for providing lubrication to the brain. CPE cells are highly secretory, and they actively produce the CSF, including nutrients and neurotropic factors (Redzic & Segal, 2004; Smith *et al*, 2004; Emerich *et al*, 2005; Abbott *et al*, 2010; Redzic, 2011). Moreover, the choroid plexus transcriptome, proteome, and secretome are dynamic and respond to inflammatory triggers in the periphery (Marques *et al*, 2007, 2009; Thouvenot *et al*, 2012; Vandenbroucke *et al*, 2012).

1   Inflammation Research Center, VIB, Ghent, Belgium
2   Department of Biomedical Molecular Biology, Ghent University, Ghent, Belgium
3   Department of Medical Protein Research, VIB, Ghent, Belgium
4   Department of Biochemistry, Ghent University, Ghent, Belgium
5   Laboratory of General Biochemistry and Physical Pharmacy, Faculty of Pharmaceutical Sciences, Ghent University, Ghent, Belgium
6   Department of Neurobiology, Institute for Biological Research, University of Belgrade, Belgrade, Republic of Serbia
7   Laboratory of Experimental Cancer Research, Department of Radiation Oncology and Experimental Cancer Research, Ghent University, Ghent, Belgium
    *Corresponding author. Tel: +32 9 3313587; Fax: +32 9 3313609; E-mail: roosmarijn.vandenbroucke@irc.vib-ugent.be
    †These authors contributed equally to this work

Extracellular vesicles (EVs), including all membrane-derived vesicles that are outside the cell, have evolved as important mediators of cell–cell communication (Akers *et al*, 2013; El Andaloussi *et al*, 2013; Raposo & Stoorvogel, 2013; Colombo *et al*, 2014; Yanez-Mo *et al*, 2015; Abels & Breakefield, 2016; Tkach & Thery, 2016). Based on their mode of biogenesis, EVs can be classified as exosomes, microvesicles, or apoptotic bodies. Exosomes (30–200 nm) are secreted via fusion of multivesicular bodies (MVBs) with the plasma membrane, whereas microvesicles (200–1,000 nm) originate by direct membrane budding (Colombo *et al*, 2014). Apoptotic bodies are much larger (0.5–3 μm) and are formed by random blebbing of the plasma membrane induced by cell death. EVs carry proteins, lipids, mRNA, miRNA, and other soluble factors to both adjacent and distant cells (Akers *et al*, 2013; El Andaloussi *et al*, 2013; Raposo & Stoorvogel, 2013; Colombo *et al*, 2014; Yanez-Mo *et al*, 2015; Abels & Breakefield, 2016; Tkach & Thery, 2016).

MiRNAs present in EVs are of particular interest because they can modulate the gene expression pattern in recipient cells. MiRNAs are single-stranded RNA molecules of 19–23 nucleotides and regulate gene expression via mRNA degradation or translational repression (Filipowicz *et al*, 2008). Disturbances in the expression pattern of miRNAs are related to various pathophysiological conditions, such as cancer, inflammation, and diabetes. The presence of miRNAs outside the cell in various biological fluids (Weber *et al*, 2010) and the mechanism of miRNA secretion and its function in intercellular cross talk have recently gained significant attention (Bang & Thum, 2012; Gutierrez-Vazquez *et al*, 2013; Record *et al*, 2013; Guay *et al*, 2015; Heusermann *et al*, 2016; Tkach & Thery, 2016), also in the CNS (Andras & Toborek, 2016; Budnik *et al*, 2016; Thompson *et al*, 2016; Zappulli *et al*, 2016). Interestingly, this cross talk can lead to either pro- or anti-inflammatory responses (Vincent-Schneider *et al*, 2002; Abusamra *et al*, 2005; Kim *et al*, 2005, 2006; Bhatnagar *et al*, 2007; Clayton *et al*, 2008), while in neurodegenerative diseases such as Parkinson's disease, Alzheimer's disease, and amyotrophic lateral sclerosis, it is believed that these EVs carry toxic proteins within the CNS (Vingtdeux *et al*, 2007; Vella *et al*, 2008; Guest *et al*, 2011; Fruhbeis *et al*, 2012; Rajendran *et al*, 2014).

Given the highly secretory nature of CPE cells and their strategic position at the interface of blood and CSF, they might play an important role in the communication between the blood and the brain. In this study, we identified a novel way of blood-to-brain communication that is activated by peripheral inflammation and occurs via secretion of CPE-derived, miRNA-containing EVs into the CSF. These CPE-derived EVs carry miRNA molecules, cross the ependymal cell layer lining the ventricles, reach the recipient brain parenchymal cells, and induce target mRNA repression and inflammatory response activation.

# Results

### Systemic inflammation induces changes in extracellular vesicles (EVs) in the cerebrospinal fluid (CSF)

Cerebrospinal fluid (CSF) occupies the subarachnoid space and the ventricular system around and inside the brain and spinal cord, serves as a shock absorber for the central nervous system (CNS) and circulates nutrients and chemicals filtered from the blood into the brain; thereby playing an important role in brain homeostasis. Next to plasma proteins, electrolytes, amino acids, etc., CSF also contains extracellular vesicles (EVs). EVs are membrane-derived vesicles, including exosomes, microvesicles, or apoptotic bodies. Indeed, transmission electron microscopy (TEM) analysis confirmed the presence of EVs in the CSF isolated from mice (Fig 1A). Interestingly, nanoparticle tracking analysis (NTA, NanoSight) measurements of the amount of particles in CSF isolated from control and LPS-injected mice revealed that the amount of EVs in CSF increases gradually upon systemic inflammation, and the rise is significant 2 h after LPS injection. Figure 1B and C displays the absolute numbers and size distributions of the particles, respectively, and indicates that systemic inflammation leads to an increased amount of EVs in the CSF, especially in EVs of ~110 nm in size.

Encapsulated in the extracellular vesicles are proteins and nucleic acids including miRNAs. To address whether the increase in EVs upon systemic inflammation was associated with an increase in miRNAs in the CSF, we pooled 50 μl CSF from different mice and isolated EVs, followed by total RNA isolation. Next, we analyzed expression levels of several miRNAs implicated in inflammation (Sheedy & O'Neill, 2008), namely miR-1a, miR-9, miR-146a, and miR-155, and all of them showed up-regulation in the CSF-derived EVs after *in vivo* systemic (intraperitoneal, i.p.) LPS injection (Fig 1D–G).

### Primary choroid plexus epithelial (CPE) cells secrete miRNA-containing EVs upon LPS treatment *in vitro*

The choroid plexus hangs in the ventricles and contains a single layer of CPE cells surrounding a core of fenestrated capillaries and loose connective tissue. These CPE cells are uniquely positioned between blood and CSF and are responsible for most of the CSF production. Here, we hypothesized that the CPE cells sense peripheral inflammation at their basal side, resulting in EV secretion at the apical side. To address this, we extended our study to the *in vitro* response of primary cultures of mouse CPE cells. We cultured primary CPE cells as described (Menheniott *et al*, 2010) and plated them onto transwells to mimic the *in vivo* situation (Fig EV1A). We thoroughly characterized the primary CPE cells by the expression of transthyretin (data not shown) and the presence and functionality of tight junctions. The primary CPE cells were strongly positive for zona occludens (ZO1, red), E-cadherin (ECDH, green), and claudin-1 (CLDN1, red) (Fig EV1B–D). Additionally, transepithelial electrical resistance (TEER) measurements confirmed the formation of a tight barrier (Fig EV1E).

The primary CPE cells were stimulated with LPS from the basal side, after which the supernatant was analyzed. Figure 2A and B displays the number and size distribution of the particles in the supernatant determined by NTA analysis (NanoSight). This revealed that LPS stimulation of primary CPE cells from the basal side results in increased secretion of EVs into the supernatant. Next, EVs were isolated, followed by RNA isolation and miRNA expression analysis. Analysis of the EV-associated miRNAs (Fig 2C–E) showed LPS-dependent miR-9, miR-146a, and miR-155 up-regulation, while miR-1a expression level was below detection limit. In parallel, we also analyzed miRNA expression of the CPE cells. qPCR analysis revealed that miR-1a/-9 were down-regulated and miR-146a/-155 were up-regulated in LPS-stimulated primary CPE cells (Fig 2F–I).

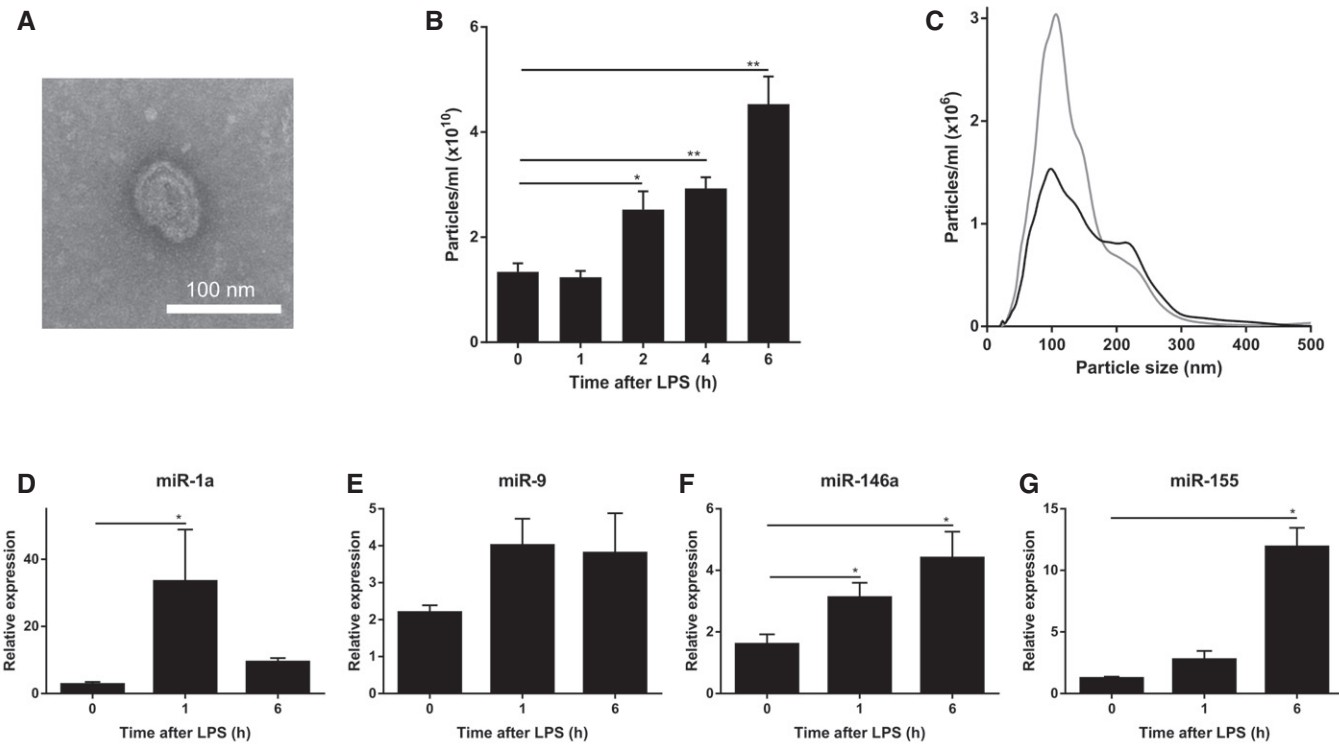

**Figure 1.  LPS injection induces changes in extracellular vesicles (EVs) and miRNAs in the cerebrospinal fluid (CSF).**

A    Representative transmission electron microscope (TEM) image showing the presence of EVs in the CSF in two independent experiments.

B    NanoSight quantification of the amount of particles in the CSF 0, 1, 2, 4, and 6 h after i.p. LPS injection ($n$ = 3–5).

C    Size distribution of the EVs *in vivo* in the CSF before (black; $n$ = 5) and 6 h after (gray; $n$ = 3) LPS treatment determined by NanoSight analysis.

D–G    Quantitative real-time polymerase chain reaction analysis of miR-1a (D), miR-9 (E), miR-146a (F), and miR-155 (G) ($n$ = 4). RNA was isolated from pooled CSF (50 μl) from different mice ($n$ = 3).

Data information: Data in (B, D-G) are displayed as mean ± SEM and analyzed by Student's $t$-test. Significance levels are indicated on the graphs: *$0.01 \leq P < 0.05$; **$0.001 \leq P < 0.01$.

This might indicate that miR-1a and miR-9 are directly secreted into the CSF without new synthesis of the miRNAs (resulting in up-regulation in supernatant and down-regulation in the CPE cells), while miR-146a and miR-155 are secreted but also their transcription is highly increased (resulting in up-regulation both in supernatant and CPE cells).

Exosomes are EVs of 30–200 nm that are secreted by exocytosis from multivesicular bodies (MVBs) and that are characterized by the presence of specific marker proteins (Mathivanan & Simpson, 2009). In agreement with the observed increase in EV secretion by the CPE cells upon LPS stimulation, we also observed increased MVB production *in vitro*. Indeed, CD63 immunostaining (red) of primary CPE cells showed large intracellular vesicles in the presence of LPS (Fig EV2B; white arrow heads), which were absent in untreated primary CPE cells (Fig EV2A). Additionally, this increase was also detected by RiboGreen staining (Ganguly *et al*, 2009; Chiba *et al*, 2012) (green) of unstimulated (Fig EV2C) and LPS-stimulated primary CPE cells (Fig EV2D).

Inhibiting the exosome-mediated miRNA secretion by addition of the neutral sphingomyelinase inhibitor GW4869, a validated inhibitor of exosome production (Trajkovic *et al*, 2008), to primary CPE cells further suggested that the observed LPS effects were exosome dependent. NanoSight analysis showed that GW4869 treatment

resulted in a reduction of LPS-induced EV secretion (Fig 3A). Interestingly, this decrease in EV secretion was correlated with a reduction in miR-9, miR-146a, and miR-155 expression in EVs isolated from the supernatant of LPS-stimulated primary CPE cells (Fig 3B–D). In contrast, exosome inhibitor treatment induced accumulation of miR-1a, miR-9 and miR-155 but not miR-146a in LPS-stimulated primary CPE cells (Fig 3E–H). These data show that CPE cells secrete miRNA-containing exosomes into the supernatant in response to LPS.

### CPE cells are the main source of miRNA-containing EVs that are released into the CSF upon systemic inflammation *in vivo*

Transthyretin (TTR) is a protein that consists of four identical subunits of 14 kDa in a tetrahedral symmetry (Ingenbleek & Young, 1994). Plasma TTR originates primarily from the liver, whereas brain TTR is exclusively produced, secreted, and regulated by the choroid plexus (Herbert *et al*, 1986; Aldred *et al*, 1995). Interestingly, Western blot analysis of EV samples isolated from CSF revealed the presence of TTR (Fig EV3). Clearly, EVs isolated from CSF contain a choroid plexus specific marker, suggesting that the choroid plexus is an important source of the EVs that are present in the CSF.

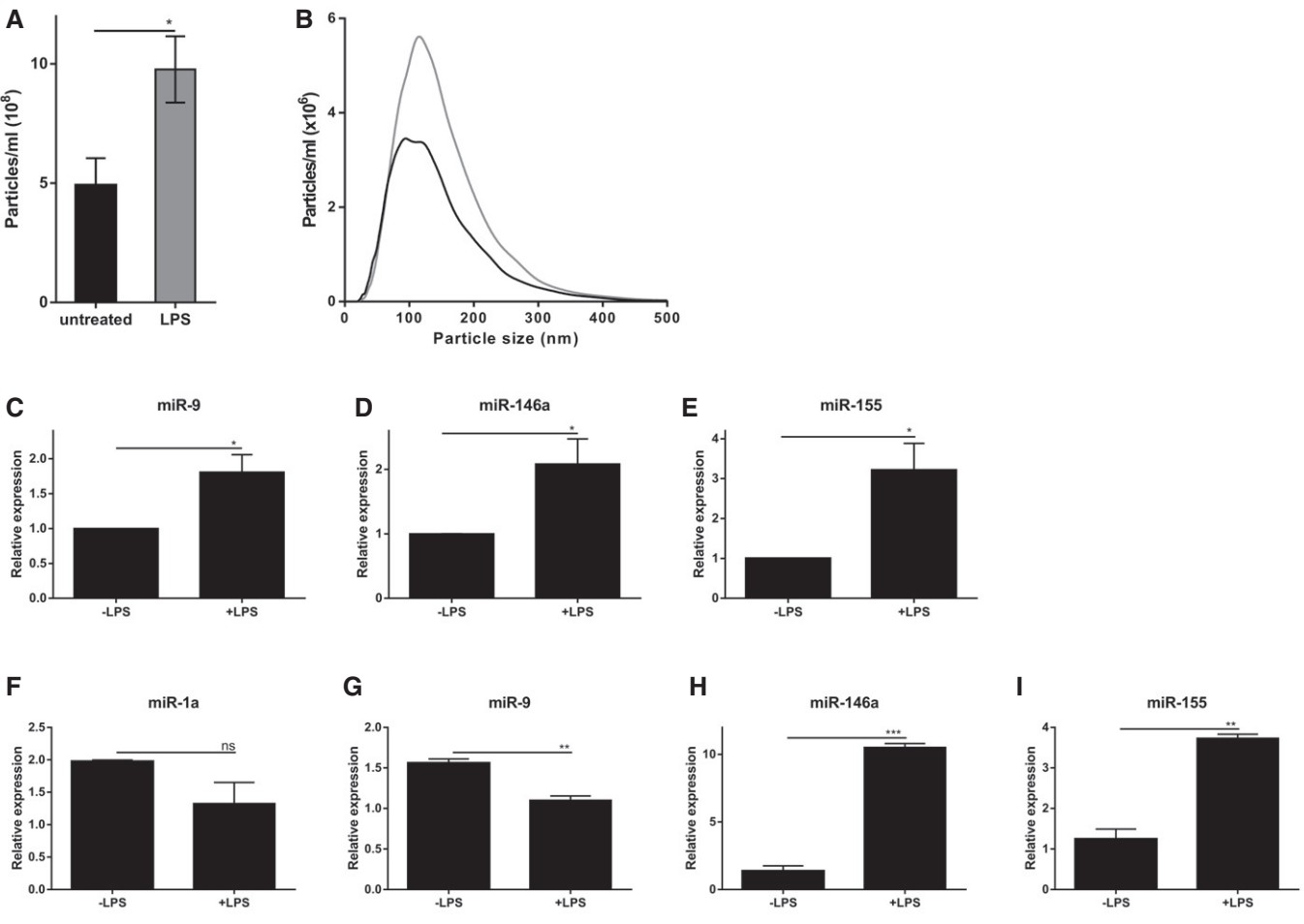

**Figure 2.  Primary choroid plexus epithelial (CPE) cells secrete miRNA-containing EVs upon LPS incubation *in vitro*.**

A, B  *In vitro* quantification (A) and size distribution (B) of EVs isolated from conditioned medium of primary CPE cells grown in a transwell system after 12 h in the absence (black) or presence (gray) of LPS (*n* = 5) determined by NanoSight analysis.

C–E  TaqMan qPCR assay for the quantification of miR-9 (C), miR-146a (D), and miR-155 (E) in the exosomal pellet isolated from conditioned medium of primary CPE cells grown in a transwell system and stimulated for 12 h with LPS (*n* = 3).

F–I  Quantification of the miRNAs miR-1a (F), miR-9 (G), miR-146a (H), and miR-155 (I) by TaqMan qPCR assay from primary CPE cells grown in a transwell system without or with LPS stimulation (*n* = 3).

Data information: Data in (A, C–I) are displayed as mean ± SEM and analyzed by Student's *t*-test. Significance levels are indicated on the graphs: *0.01 ≤ *P* < 0.05; **0.001 ≤ *P* < 0.01; ***0.0001 ≤ *P* < 0.001.

To study whether CPE cells show changes in EV production upon i.p. LPS injection *in vivo*, similar to what we observed *in vitro*, we analyzed choroid plexus gene expression of different EV markers. Several of the tested markers were strongly altered in the CPE cells *in vivo* after 6 h LPS treatment: *Hspa1a*, *Cd63*, and *Anxa5* were up-regulated while *Cd9* and *Cd81* were down-regulated (Fig EV4A–E, indicating an effect on the exosome machinery). Furthermore, we performed immunofluorescence analysis of different EV markers, namely CD63, RAB5, and ANXA2, on brain sections of naive mice and 4 and 8 h after LPS injection. This revealed a strong induction of all tested EV proteins early upon stimulation with LPS (Fig 4A). CD63 was mainly observed in the perinuclear area in basal conditions and early upon LPS stimulation there is an increased signal at the apical side, close to the CSF. At a later time point, high CD63 levels are observed both at the perinuclear area and at the apical side of the choroid plexus epithelial cells. Similarly, RAB5 can be

detected in the choroid plexus of naive mice and LPS stimulation results in higher levels of RAB5 both in the cytoplasm and at the apical side of the choroid plexus epithelial cells. Although ANXA2 expression was less homogeneous throughout the choroid plexus, this marker is expressed at basal conditions and is strongly induced upon LPS stimulation.

Moreover, TEM of the choroid plexus revealed a huge increase in amount of exosomes in the MVBs of LPS-treated mice (Fig 4C) compared to MVBs in the choroid plexus of unchallenged mice (Fig 4B). We quantified both the amount of MVBs per cell and the amount of exosomes per MVB at different time points. Figure EV4F–K shows representative TEM images of the choroid plexus in the absence of LPS and 1, 2, 3, 4, and 6 h after peripheral LPS injection. In Fig 4D and E, the amount of MVBs per cell section and the average amount of exosomes per MVB, respectively, were quantified. Additionally, we calculated the total

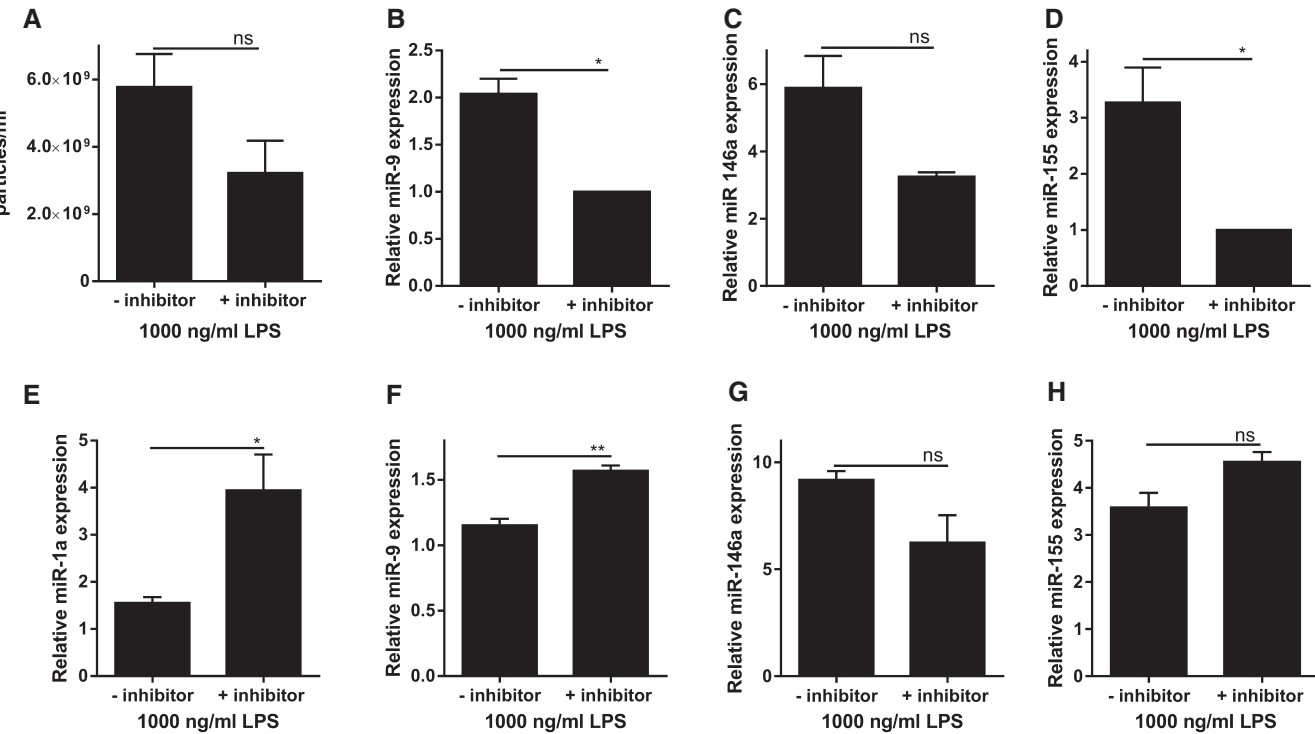

**Figure 3.  Effect of exosome inhibition on EV and miRNA secretion of primary CPE cells stimulated with LPS.**

A    *In vitro* quantification of EVs isolated from conditioned medium of LPS-stimulated primary CPE cells grown in a transwell system in the absence or presence of the exosome inhibitor GW4869 (*n* = 3).

B–D    TaqMan assay quantification of the miRNAs miR-9 (B), miR-146a (C), and miR-155 (D) in supernatant of LPS-stimulated primary CPE cells grown in a transwell system and either left untreated or pretreated with GW4869 to inhibit exosome secretion (*n* = 3). miR-1a levels were below detection limit.

E–H    TaqMan assay quantification of the miRNAs miR-1a (E), miR-9 (F), miR-146a (G), and miR-155 (H) in cell lysates of LPS-stimulated primary CPE cells grown in a transwell system left untreated or treated with GW4869 to inhibit exosome secretion (*n* = 3).

Data information: Data are displayed as mean ± SEM and analyzed by Student's *t*-test. Significance levels are indicated on the graphs: *0.01 ≤ *P* < 0.05; **0.001 ≤ *P* < 0.01.

amount of exosomes per cell section (Fig 4F) and found that LPS induces an increase in the amount of MVBs per cell and in the amount of exosomes per MVB. Four and three hours after LPS injection, the amount of MVBs per cell and the amount of exosomes per MVB reached a maximum, respectively. These kinetics resemble the EV kinetics in CSF quantified by NanoSight analysis as described above (Fig 1B) and provide evidence that the CPE cells are responsible for the observed increase of EVs in CSF upon i.p. LPS injection.

Next, we studied whether the different miRNAs that were detected in the CSF, could also be detected in the CPE cells *in vivo*. In situ hybridization (ISH) of miR-146a, miR-9, and miR-155 revealed that the miRNAs were present in the CPE cells (Fig EV4L–N), while ISH of miR-1 was unsuccessful. However, all miRs could be detected by qPCR and Fig 4G–J displays the kinetic expression profile of miR-1a, miR-9, miR-146a, and miR-155 upon LPS i.p injection in choroid plexus tissue. MiR-1a and miR-9 were significantly down-regulated upon systemic inflammation, while miR-146a and miR-155 were up-regulated. This is in agreement with what we observed *in vitro* in the primary CPE cells (Fig 2F–I).

Similarly to our *in vitro* experiments, we also tested the exosome inhibitor GW4869 *in vivo*. Mice were injected i.p. with

LPS and 4 h later injected intracerebroventricularly (icv) with GW4869 or vehicle. After 2 h, CSF and choroid plexus were isolated and analyzed. CSF analysis by NanoSight revealed that inhibition of exosome production reduced the amount of EVs in the CSF (Fig 4K). Moreover, this resulted in accumulation of several miRNAs in the choroid plexus. This accumulation was significant for miR-146a, miR-155, and miR-9, but not for miR-1a (Fig 4L). These results show that blocking exosome production by icv injection of an exosome inhibitor prevents exosome release from the choroid plexus and leads to miRNA accumulation in the choroid plexus.

Additionally, we performed an *ex vivo* experiment by injecting mice with PBS or LPS and 2.5 h later isolating choroid plexus after transcardial perfusion with DMEM medium to remove all blood from the vascularized choroid plexus. Isolated choroid plexus explants were kept in culture for 2.5 h in OptiMEM medium, and supernatant was analyzed by NanoSight. This analysis revealed the presence of significantly more EVs in the supernatant of choroid plexus from LPS-injected mice compared to PBS controls (Fig 4M), further providing evidence that the choroid plexus is the main source of the observed LPS-dependent increase in EVs in the CSF.

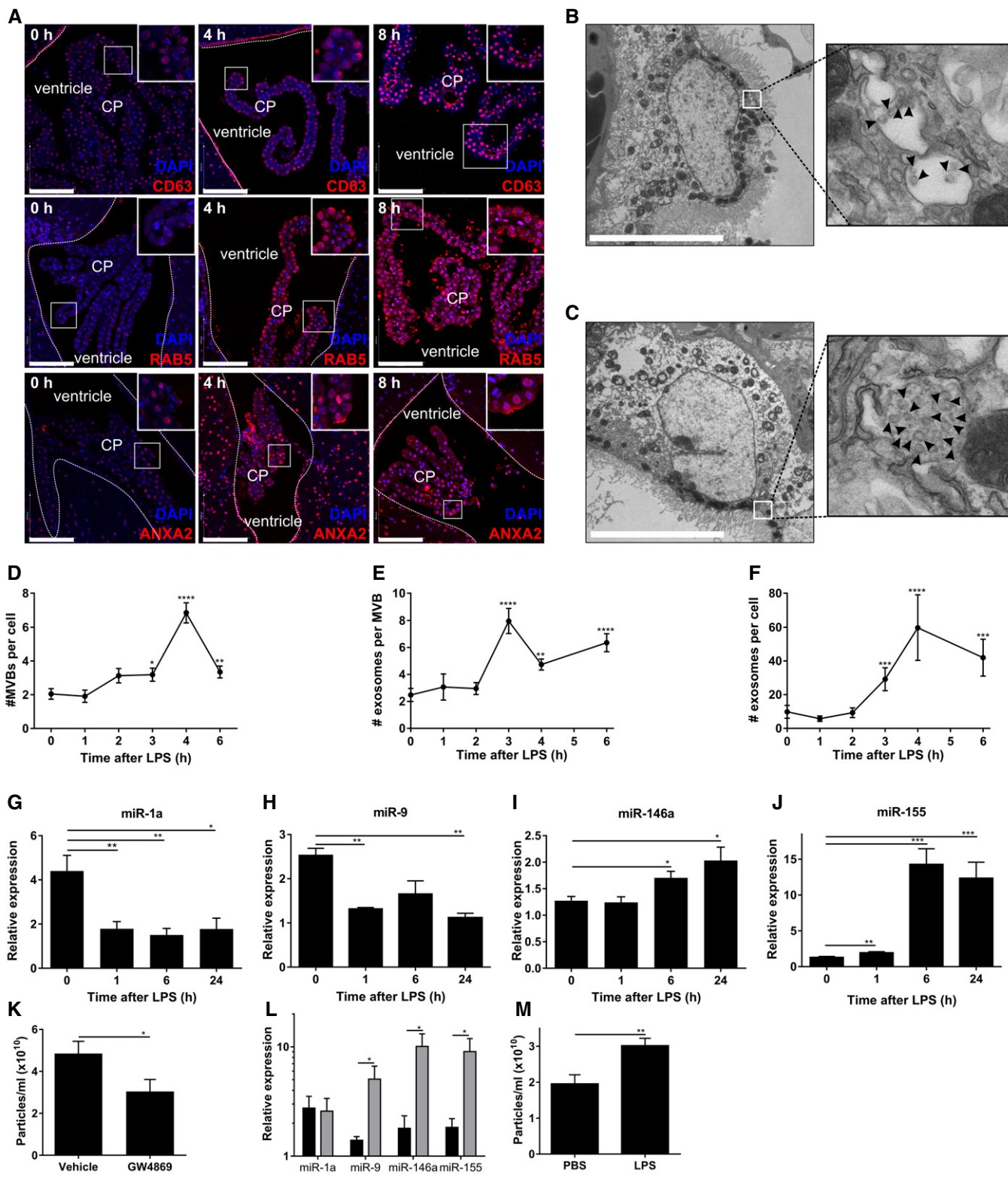

**Figure 4.**

**Different peripheral stimuli induce miRNA-containing EVs that are released into the CSF *in vivo***

Next, we studied whether CPE cells have the ability to sense peripheral inflammation by analyzing Toll-like receptor 4 (TLR4)

and tumor necrosis factor 1 (TNFR1) levels, two important receptors involved in inflammatory signaling. Immunofluorescence analysis of TLR4 and TNFR1 in choroid plexus of naive and LPS-treated mice revealed that both TLR4 (Fig 5A) and TNFR1 (Fig 5B) are expressed in CPE cells and both are up-regulated upon LPS

**Figure 4.   Systemic inflammation activates the exosomal machinery in the choroid plexus.**

A   Representative confocal images of CD63, RAB5, and ANXA2 (red) in the choroid plexus (CP) at 0, 4, and 8 h after LPS treatment. Hoechst (blue) was used to stain the nucleus. The dotted line indicates the ependymal cells that line the ventricle and the square boxes indicate the zoomed insert images displayed at the right corner of each image. Scale bars, 100 μm.

B, C   Representative TEM images showing the presence of MVBs in the CPE cells before (B) and 6 h after (C) LPS administration *in vivo*. Black arrow heads point to exosomes present in MVBs. Scale bars, 9 μm.

D–F   Quantification of number of MVBs per cell section (D), number of exosomes per MVB (E), and number of exosomes per cell section (F), based on TEM analysis of several adjacent cells (0 h, $n = 20$; 3 h, $n = 21$; 4 h, $n = 13$; 6 h, $n = 23$).

G–J   Quantitative real-time polymerase chain reaction (qPCR) analysis of miR-1a (G), miR-9 (H), miR-146a (I), and miR-155 (J). Data are presented as relative expression normalized with housekeeping miRs by TaqMan qPCR assay (0 h, $n = 4$; 1 h, $n = 5$; 6 h, $n = 5$; 24 h, $n = 3$).

K   NanoSight analysis of CSF isolated from LPS-injected mice followed by icv injection of vehicle or GW4869, a neutral sphingomyelinase inhibitor that inhibits exosome secretion ($n = 8$).

L   qPCR analysis of the expression of miR-1a, miR-9, miR-146a, and miR-155 in the choroid plexus of mice injected with LPS and then icv injected with vehicle (black) or GW4869 (gray) ($n = 4$).

M   NanoSight analysis of the supernatant of choroid plexus explants from PBS- or LPS-injected mice ($n = 6$).

Data information: Data in (D–M) are displayed as mean ± SEM and analyzed by Student's *t*-test. Significance levels are indicated on the graphs: *$0.01 \leq P < 0.05$; **$0.001 \leq P < 0.01$; ***$0.0001 \leq P < 0.001$; ****$P < 0.0001$.

injection, suggesting that the CPE cells have immune sensing capabilities.

Additionally, we analyzed whether LPS-induced EV secretion by the CPE cells also occurs after inducing peripheral inflammation by the cecal ligation and puncture (CLP) model, the most widely used mouse model of human sepsis (Dejager *et al*, 2011). Figure 5C–E reveals that CLP induced a significant increase in the amount of EVs in CSF (Fig 5C) and miRNA changes in the choroid plexus (Fig 5D) and CSF (Fig 5E) resembling those after systemic LPS injection.

To study whether cytokine production is responsible for the observed changes in EV and miRNA production at the CPE, we injected mice with recombinant TNF (25 μg/20 g body weight) and studied EV and miRNA changes in both CSF and choroid plexus. Systemic TNF induced an increase in total amount of vesicles in the CSF (Fig 5F). Additionally, miRNA analysis of the choroid plexus revealed a significant increase in both miR-155 and miR-146a, while miR-9 is decreased and miR-1a is not affected (Fig 5G). CSF miRNA expression analysis revealed up-regulation of two of the tested miRs: miR-155 and miR-146a and miR-9 and miR-1a were unaffected (Fig 5H). This shows that TNF is responsible for at least a part of the effects that are induced by systemic LPS injection or CLP.

**Analysis of the EV proteome content reveals changes upon systemic inflammation**

Proteomic studies have yielded extensive lists of proteins that are detected in different types of EVs. Here, we purified EVs from CSF and performed a label-free proteome analysis (Appendix Fig S1A). EVs were isolated from 50 μl of CSF pooled from mice injected with PBS or LPS, shortly separated by SDS–PAGE, in-gel digested with trypsin, pre-fractionated by RP-HPLC, and analyzed by LC-MS/MS. The proteins were identified using the Mascot algorithm, and MaxLFQ was used for quantification (Cox *et al*, 2014). We identified about 1,000 unique proteins in each EV sample, which is comparable to a study in which proteome analysis of human CSF EVs, isolated by ultracentrifugation, was optimized (Chiasserini *et al*, 2014). Reproducibility of the proteomics was high, reflected by the high Pearson correlation coefficients of the comparison of the LFQ intensities of the different samples (Appendix Fig S1B) and the high overlap (> 80%) in identified proteins between the replicates both for EV samples from control mice (Appendix Fig S1C) and from LPS-injected mice (Appendix Fig S1D). Comparison of the proteomes of EVs in CSF of control and LPS-injected mice, taking into account only the proteins detected in both replicates, reveals that 107 and 280 proteins are specific for the control and LPS-injected mice, respectively (Fig 6A). The EV proteome contained 14 out of the top 25 exosome markers published on the ExoCarta website (http://www.exocarta.org/) (Kalra *et al*, 2012). Figure 6B shows the overlap between the exosomal proteins from the EVpedia database (Kim *et al*, 2013) (http://evpedia.info; *Mus musculus in vitro/in vivo*), and the proteins present in both EV samples derived from control and LPS-injected mice; the overlap is ~30%. It is important to realize that the available

**Figure 5.   The choroid plexus responds to different systemic inflammatory triggers.**

A, B   Representative confocal images of choroid plexus (CP) on brain sections from naive mice and 4 h after LPS injection ($n = 3$). Brain sections were stained for TLR4 (A) and TNFR1 (B) (red), pan-cytokeratin (panCK, green), and the nuclei were stained with Hoechst (blue). The ependymal cells lining the ventricles are marked with a dotted line. Scale bars, 100 μm.

C   NanoSight analysis of EVs in the CSF in sham-operated (black; $n = 3$) relative to CLP-treated (gray; $n = 4$) mice 10 h after surgery ($n = 4$).

D   qPCR analysis of the expression of miR-1a, miR-9, miR-146a, and miR-155 in the choroid plexus of sham-operated mice (black) and mice subjected to CLP (gray) 10 h after surgery ($n = 5$–7).

E   qPCR analysis of the expression of miR-1a, miR-9, miR-146a, and miR-155 in the CSF of sham-operated mice (black) and mice subjected to CLP (gray) 10 h after surgery ($n = 4$–5).

F   NanoSight analysis of EVs in CSF of control (black; $n = 5$) and TNF-injected (25 μg/20 g; gray; $n = 4$) mice 6 h after TNF challenge.

G   qPCR analysis of the expression of miR-1a, miR-9, miR-146a, and miR-155 in the choroid plexus of control mice (black) and in mice injected with TNF (gray) 6 h after injection ($n = 5$–7).

H   qPCR analysis of the expression of miR-1a, miR-9, miR-146a and miR-155 in the CSF of control mice (black) and on mice 6 h after TNF injection (gray)  ($n = 3$).

Data information: Data in (C–H) are displayed as mean ± SEM and analyzed by Student's *t*-test. Scale bar, 100 μm. Significance levels are indicated on the graphs:
*$0.01 \leq P < 0.05$; **$0.001 \leq P < 0.01$.

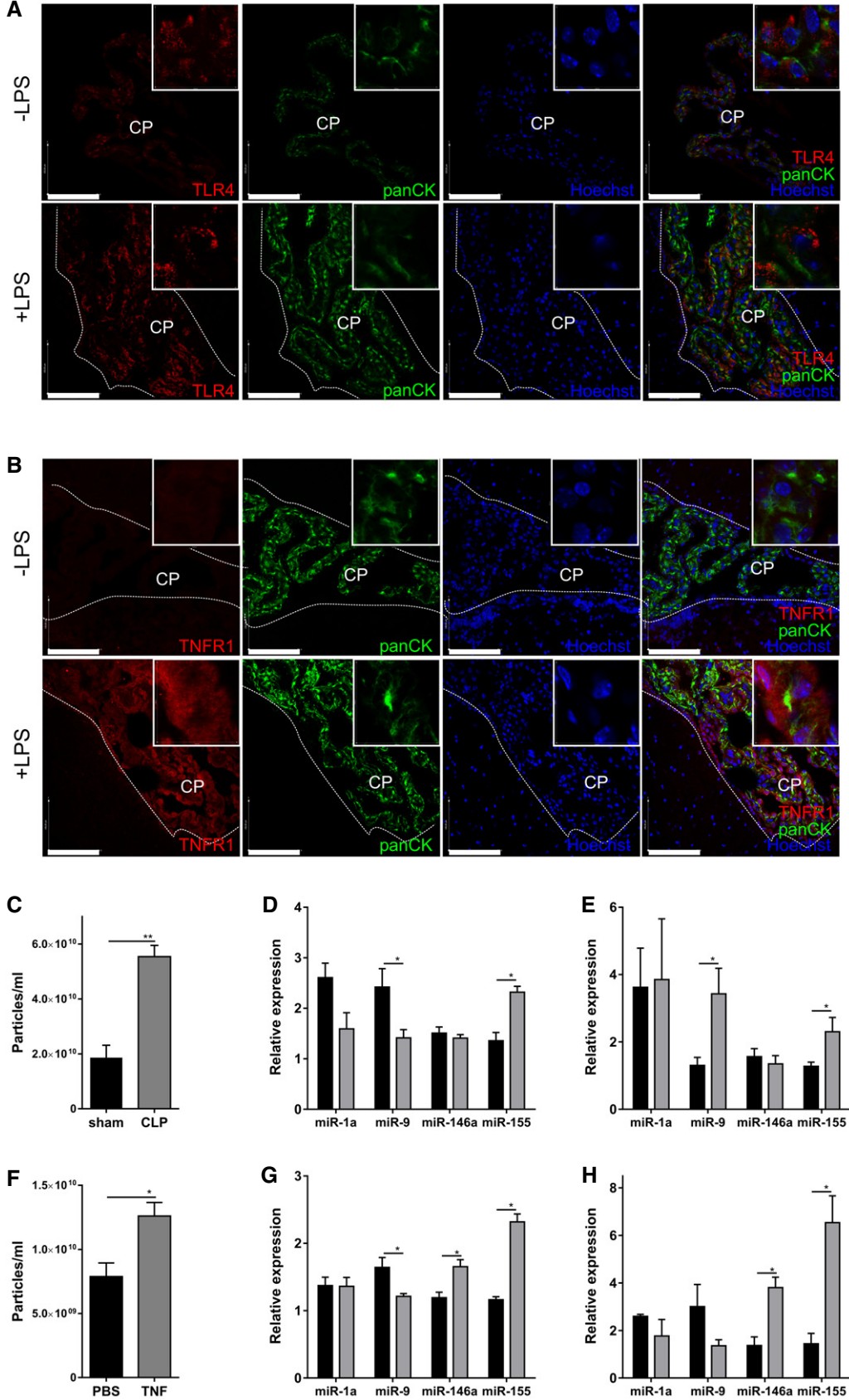

**Figure 5.**

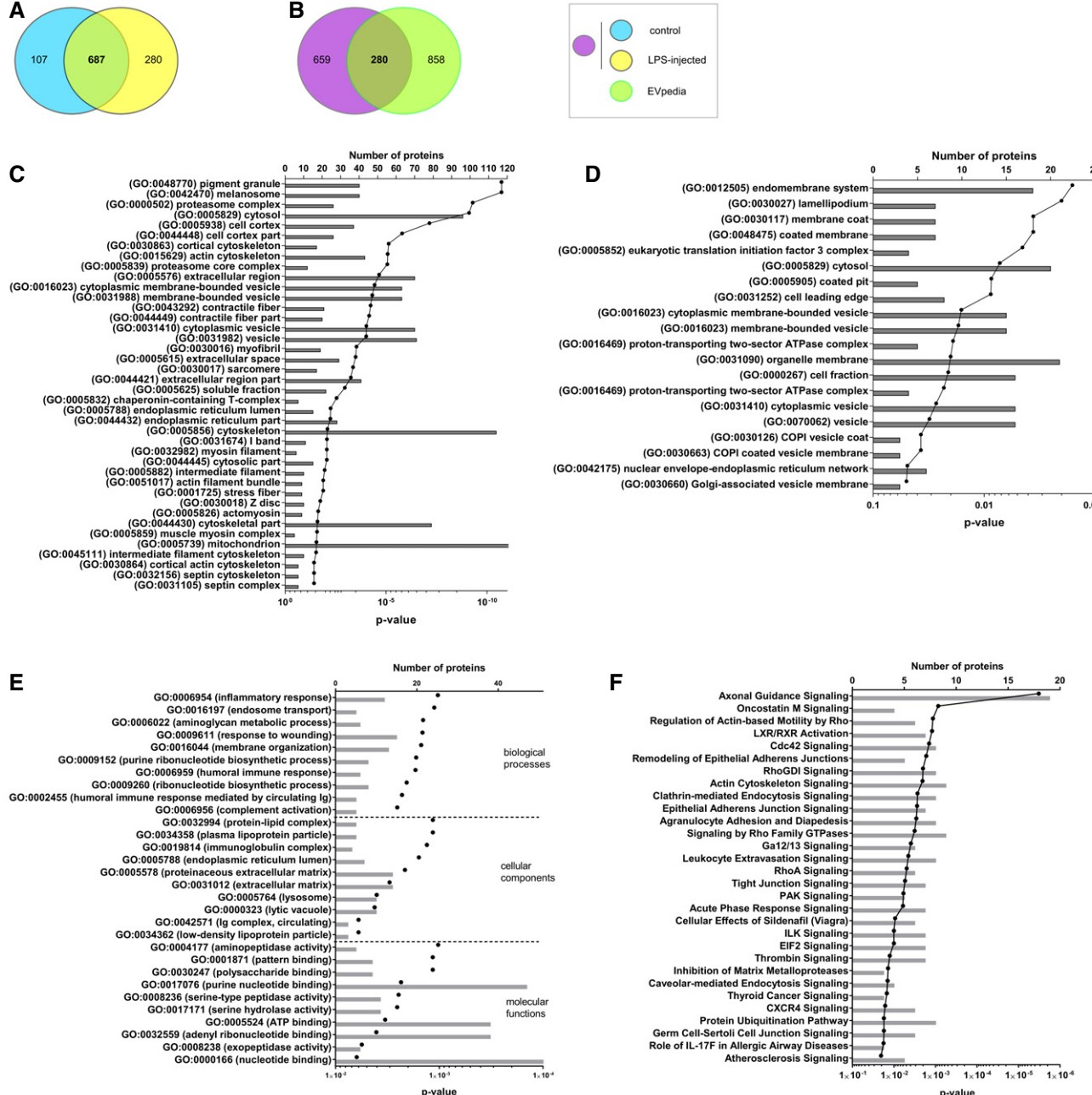

**Figure 6. Proteome analysis of CPE-derived EVs in the presence and absence of LPS.**

A   Venn diagrams showing overlap of proteins identified in EVs isolated from CSF of control (blue) and LPS-injected (yellow) mice that were present in the two independent replicates.

B   Venn diagrams showing overlap of the CSF EV proteome (control and LPS; purple) with the *Mus musculus* proteome list available on the EVpedia website (green).

C   GO enrichment analysis using DAVID of the EV proteome compared with total choroid plexus proteome.

D   GO enrichment analysis using DAVID of the EV proteome following LPS treatment compared with total choroid plexus proteome. The gray bars represent the number of proteins and the black dots the *P*-values.

E, F   DAVID (E) and Ingenuity pathway analysis (IPA) (F) on the CSF-derived EV proteome, taking into account all proteins that are exclusively detected after LPS stimulation in both replicates.

Data information: (C–F) The gray bars represent the number of proteins and the black dots the *P*-values.

databases do not contain proteome data from mouse CSF samples, which can explain the, at first sight, apparent limited overlap.

To identify the pathways enriched in the proteome of the EV samples, we performed a Gene Ontologies (GO) enrichment analysis using DAVID (da Huang *et al*, 2009). We submitted the total choroid

plexus proteome, obtained in a similar way as the EV proteome (Appendix Fig S1E), as "background list" and the EV proteome as "query list". As shown in Fig 6C, DAVID analysis reveals that "vesicular transport" is enriched in the EV proteome analysis, confirming the successful isolation of EVs. The bars represent the number of proteins and the connected dots the *P*-values. The enriched GO terms include membrane-bounded vesicles, cytoplasmic membrane-bound vesicles, and cytoplasmic vesicles. The enriched GO terms related to vesicles were even more pronounced when the "query list" was replaced by the proteins that were identified in EVs isolated from LPS-injected mice (Fig 6D). This DAVID analysis revealed the significant enrichment of endomembrane system, coated membrane, membrane coat, COPI-coated vesicle membrane, COPI vesicle coat, Golgi-associated vesicle membrane, vesicular fraction, etc. These results clearly show that we enriched for EVs and that EV release into the CSF is increased in response to systemic LPS injection.

To predict the pathways that might be affected in target cells after uptake of the LPS-induced EVs, we performed a DAVID (da Huang *et al*, 2009) analysis (Fig 6E) and an Ingenuity pathway analysis (IPA) (Fig 6F) on the 280 proteins that were specific for the EV proteome after LPS treatment, as shown in Fig 6A. Figure 6E displays the top 10 biological processes, cellular components, and molecular functions, as determined by DAVID analysis, and Fig 6F shows the top 30 canonical signaling pathways, as determined by IPA analysis. These results clearly indicate that the proteome of EVs derived from LPS-injected mice is pro-inflammatory. Moreover, DAVID analysis revealed that several proteins associated with EVs have enzymatic activity as well as nucleotide binding functions. As expected, and in agreement with the GO enrichment analysis using DAVID (Fig 6C), several proteins involved in endosomal transport and membrane organization were also enriched in the EVs isolated from CSF of LPS-injected mice. IPA analysis revealed that the proteome present in EVs isolated from LPS-injected mice might influence several processes in target cells, such as axonal guidance (by which neurons send out axons to the correct targets) and leukocyte extravasation. These data indicate that recipient cells will not only be influenced by the EV-associated miRs, but without any doubt also by the EV proteome content.

## EVs induced by systemic inflammation transfer signals to the brain parenchyma

Next, we investigated whether EVs released into the CSF can cross the ependymal cell layer and whether they are taken up by brain parenchymal cells. EVs were purified from CSF from LPS-injected mice, double-labeled with the membrane label PKH26 (red) and RNA label RiboGreen (green), to confirm the presence of the RNA inside the EVs (Appendix Fig S2A and B). Next, we injected PKH26 labeled EVs into the lateral ventricles of naive mice. Analysis of brain sections 4 h later revealed that EVs are able to cross the ependymal cell layer and penetrate into the brain parenchyma (Fig 7A). Pan-cytokeratin (panCK) was used as a marker for the CPE cells.

To determine which brain cells can take up EVs, we incubated primary mixed cortical cultures originating from neonatal mice for 4 h at 37°C with double-labeled EVs obtained from LPS challenged mice. Then, they were fixed and analyzed by confocal microscopy, which revealed that EVs are taken up by both GFAP-positive astrocytes (Fig 7B) and IBA1-positive microglia (Fig 7C), but not by TUBB3-positive neurons (data not shown). Also *in vivo*, we observed that the EVs were closely associated with GFAP-labeled cells (Fig 7A, zoom).

Next, we analyzed the effect of the EVs on gene expression of the recipient cells. Based on miRTarBase, miRecords, and miRWalk databases, we selected 10 genes that are described as mRNA targets of miR-1a, miR-9, miR-146a, and/or miR-155, namely *Mapk3*, *Notch1*, *Dicer1*, *Tab2*, *Sox2*, *Calm2*, *Smad2*, *Smad5*, *Dnmt3a,* and *Irak1*. To study whether EV-associated miRNAs can down-regulate some of their targets, we incubated primary mixed cortical cultures with EVs isolated from unchallenged mice and from mice 6 h after LPS challenge and analyzed gene expression 24 h later (Fig 8A). Almost all genes were down-regulated as expected, and this down-regulation was significant for *Mapk3*, *Notch1*, *Dicer1*, *Tab2*, *Sox2*, *Smad2*, *Dnmt3a,* and *Irak1*. Interestingly, *in vivo* gene expression analysis of brain tissue of unchallenged and LPS challenged mice revealed similar results for the tested genes (Fig 8B). All genes were down-regulated to some degree, and this down-regulation was significant for *Notch1*, *Dicer1*, *Smad2*, *Smad5,* and *Dnmt3a*. To determine whether the effect of the EVs was pro- or anti-inflammatory, we also analyzed inflammatory genes and cytokine secretion *in vitro* (in primary mixed cortical cultures and supernatant), and *in vivo* (in brain tissue and CSF in the presence and the absence of LPS). Primary mixed cortical cultures showed strong expression of *Il1b*, *Tnf*, *Il6*, *Nos2,* and *Nfkbia* after incubation with EVs isolated from challenged mice (gray), in contrast to EVs isolated from naive mice (Fig 8C, left). In agreement with this, IL-6, IL-1β, and TNF protein secretion in the supernatant was increased in the presence of EVs isolated from LPS-injected mice (Fig 8D, left). To exclude the possibility that the pro-inflammatory effects of the EVs were due to LPS contamination, we incubated EVs from LPS-injected mice with primary mixed cortical cultures derived from wild-type and TLR4$^{-/-}$ mice. No difference in induction of inflammatory response was observed (Appendix Fig S3). Gene expression analysis of brain tissue and cytokine analysis of CSF from unchallenged

**Figure 7.  EVs can cross the ependymal layer and deliver the miRNA cargo to the brain.**

A    Representative confocal image of brain parenchyma 4 h after intracerebroventricular injection of PKH26-labeled EVs (red). Astrocytes are stained with GFAP (white), nuclei with Hoechst (blue), and pan-cytokeratin (panCK, green) was used to label the choroid plexus. The dotted line shows the ventricular border and the white arrow heads point to EVs that crossed the ependymal cell layer.

B, C  Representative confocal images showing the uptake of double-labeled EVs (RiboGreen; green, PKH26, red) by astrocytes stained with GFAP (white, B) and microglia cells stained with IBA1 (white, C) incubated on mixed cortical cultures. Boxed areas are shown as zoomed images on the right and the white arrow heads point to EVs. Cell nuclei are stained with Hoechst (blue). Scale bar (B, left) = 11 μm; (B, right) = 1.7 μm and (C) = 24 μm.

Data information: The *in vivo* and *in vitro* uptake experiments were performed three and two times (*n* = 3), respectively.

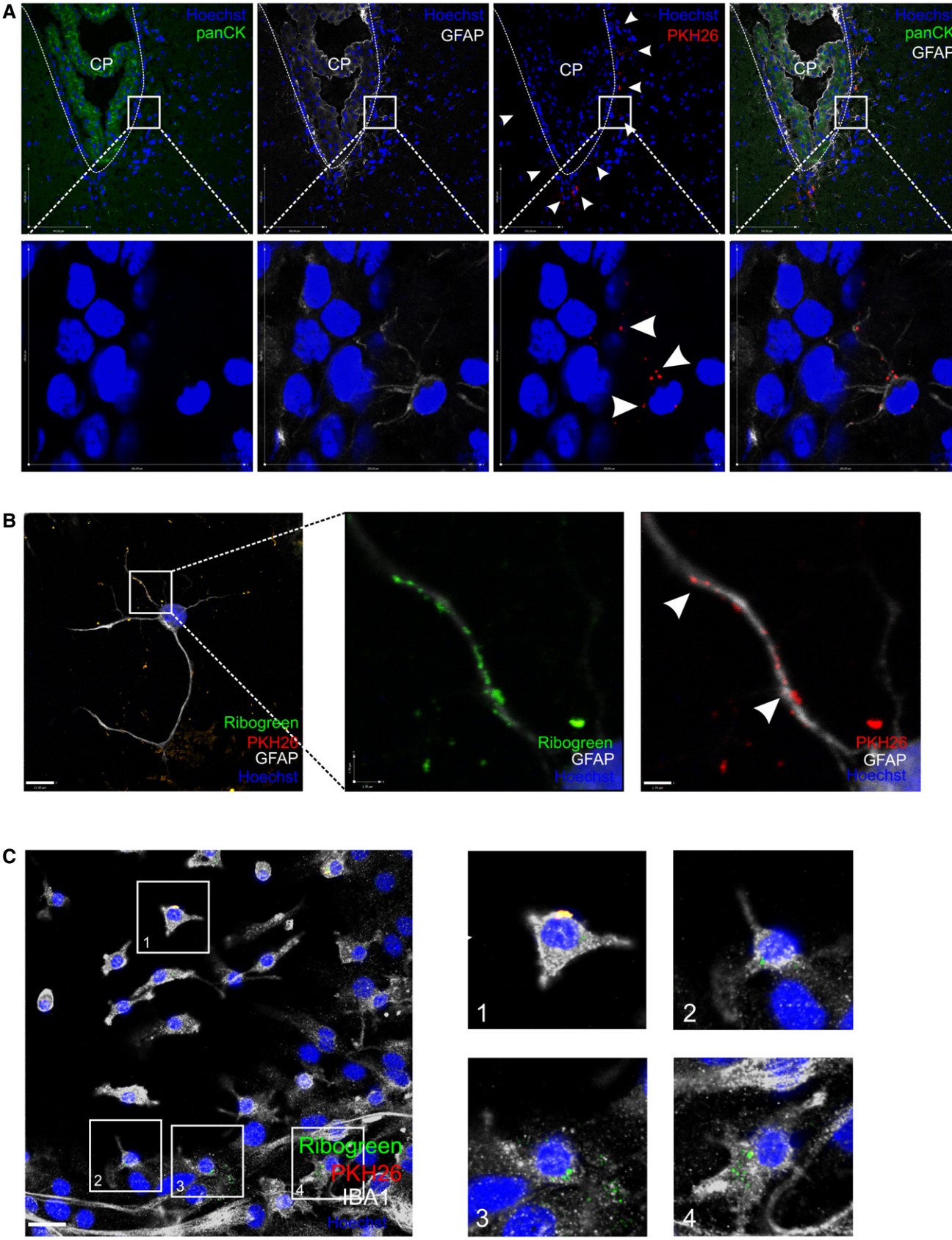

**Figure 7.**

and LPS-injected wild-type mice were in agreement with the observations that were made in the mixed cortical cultures *in vitro*: LPS resulted in up-regulation of *Il1b*, *Tnf*, *Il6*, *Nos2* and *Nfkbia* gene expression in the brain (Fig 8C, right) and increased levels of IL-6, IL-1β and TNF in the CSF (Fig 8D, right).

Next, we studied whether the observed effects of LPS-induced EVs on recipient cells were due to EVs, which are known to contain the exosomal marker CD63. Again, we isolated EVs from unchallenged and LPS-injected mice, but we removed CD63-containing EVs by using anti-CD63-coated beads. The CD63-depleted fraction was incubated on wild-type mixed cortical cultures, and inflammatory genes and miRNA target genes were analyzed (Fig 8E–F). *Nfkbia* showed a limited up-regulation, but neither *Il1b*, *Tnf*, *Il6,* nor *Nos2* were up-regulated by the CD63-depleted EV fraction (Fig 8E). Also, CD63-depleted EVs did not induce down-regulation of *Mapk3*, *Notch1*, *Dicer1*, *Tab2*, *Sox2*, *Calm2*, *Smad2*, *Smad5*, *Dnmt3a,* and *Irak1* (Fig 8F), in contrast to what was observed previously with CD63-containing EVs. All together, these results indicate that the observed effects on the recipient cells are indeed exosome dependent. Furthermore, to study the involvement of CNS barrier leakage, we analyzed the integrity of both the blood–brain barrier (BBB) and the BCSFB. Four hours after LPS injection, no leakage of 4 kDa could be observed at the BBB, indicating that the BBB is still intact at this time point (Fig EV5A). In contrast, as we have shown before (Vandenbroucke *et al*, 2012), the BCSFB integrity is affected early upon LPS injection. However, while there was an increase in leakage of intravenous (i.v.) injected 4 kDa FITC-dextran into the CSF (Fig EV5B), this was absent in case of 20 kDa FITC-dextran (Fig EV5C), indicating that the barrier disturbance is limited at this time point.

Finally, we investigated *in vivo* whether blockage of EV secretion by the CPE cells results in reduced inflammation in the brain. Similarly as described above, mice were injected i.p. with LPS and 4 h later icv with GW4869 or vehicle. Two hours later, brain tissue was isolated and analyzed. In agreement with the results obtained *in vitro*, qPCR analysis of brain tissue revealed that inhibition of inflammation-induced exosome production results in up-regulation of miR target (Fig 8G) and down-regulation of inflammatory genes (Fig 8H) in the brain; this was significant for *Dicer1*, *Tab2*, *Sox2*, *Dnmt3a*, *Irak2*, *Il1b*, *Tnf*, *Il6*, *Nos2,* and *Nfkbia*. These data provide evidence that the peripheral inflammation-induced, choroid plexus-derived exosome secretion propagates a pro-inflammatory message to the brain.

## Discussion

Knowledge about how the periphery communicates with the CNS during systemic inflammation remains limited. Three tight barriers separate the brain from the periphery to prevent peripheral blood fluctuations from disturbing brain homeostasis. These barriers are needed to assure a balanced and well-controlled microenvironment around synapses and axons to allow signaling in the CNS. One of these barriers, the BCSFB, is composed of a single layer of CPE cells that form an interface between the blood and the CSF of the brain. The choroid plexus is not only a barrier, but also a site of active protein synthesis that possesses various receptors for molecules involved in the inflammatory process. Consequently, we and others believe that these cells are uniquely positioned at the interface of the blood and the brain to act as a brain "immune sensor" involved in sensing signals from the periphery and communicate about these signals to the CSF and the brain (Vandenbroucke *et al*, 2012; Lehtinen *et al*, 2013). Here, we show that CPE cells indeed perform that function during systemic inflammation, via the secretion of EVs.

LPS is a component of the outer cell wall of Gram-negative bacteria, binds to its receptor TLR4, and induces a very robust inflammatory response when injected in mammals (Beutler & Poltorak, 2001a,b). The inflammatory profile induced by LPS is similar to that in sepsis, a hugely unmet clinical need which involves inflammation of the brain (Vandenbroucke *et al*, 2012; Maclullich *et al*, 2013). TEM analysis revealed that the CSF contained EVs of ~100 nm, which are known to contain miRNA and to be important mediators of cell–cell communication (Akers *et al*, 2013; El Andaloussi *et al*, 2013; Raposo & Stoorvogel, 2013; Colombo *et al*, 2014; Yanez-Mo *et al*, 2015). Interestingly, i.p. injection of LPS followed by CSF analysis revealed that LPS induces an increase in the total amount of EVs in the CSF. Moreover, four inflammatory miRNAs, namely miR-1a, miR-9, miR-146a, and miR-155 (Lindsay, 2008; O'Neill *et al*, 2011) were up-regulated in the EVs isolated from CSF upon inflammation. In contrast, miR-1a and miR-9 were down-regulated in the CPE and miR-146a and miR-155 were up-regulated. All these miRNA changes, except for miR-1a in the CSF due to detection limitations, were confirmed in primary CPE cultures that were stimulated *in vitro* with LPS. This suggests that some miRNAs are directly secreted into the CSF, while other miRNAs are both actively transcribed and secreted upon LPS stimulation. The latter results in up-regulation both in CSF and choroid plexus. These results suggest

---

**Figure 8. EVs repress mRNAs in target cells and transfer a pro-inflammatory message that is exosome dependent.**

A, B qPCR gene expression analysis of miRNA target genes *Mapk3*, *Notch1*, *Dicer1*, *Tab2*, *Sox2*, *Calm2*, *Smad2*, *Smad5*, *Dnmt3a*, and *Irak1* in mixed cortical cultures incubated with EVs isolated from CSF from untreated (black) or LPS-treated (gray) mice (A) and in brain tissue (B) before (black) and 8 h after LPS injection (gray) (*n* = 3).

C qPCR gene expression analysis of inflammatory genes *Il1b*, *Tnf*, *Il6*, *Nos2*, and *Nfkbia* by qPCR in mixed cortical cultures (left, *in vitro*) incubated with EVs isolated from CSF from untreated (black) or LPS-treated (gray) mice and in brain tissue (right, *in vivo*) before (black) and 8 h after LPS injection (gray) (*n* = 3).

D Cytokine analysis (IL-6, IL-1β, and TNF) of supernatant of mixed cortical cultures (left, *in vitro*) incubated with EVs isolated from untreated (black, *n* = 3) or LPS-treated (gray; *n* = 3) mice and of CSF (right, *in vivo*) from untreated (black; *n* = 3) and LPS-treated mice (gray; *n* = 6).

E, F qPCR gene expression analysis of inflammatory genes *Il1b*, *Tnf*, *Il6*, *Nos2*, and *Nfkbia* (E) and miRNA target genes *Mapk3*, *Notch1*, *Dicer1*, *Tab2*, *Sox2*, *Calm2*, *Smad2*, *Smad5*, *Dnmt3a*, and *Irak1* (F) in mixed cortical cultures incubated with CD63-depleted EVs isolated from untreated mice (black) and mice treated with LPS for 6 h (gray) (*n* = 3).

G, H qPCR gene expression analysis of miRNA target genes *Mapk3*, *Notch1*, *Dicer1*, *Tab2*, *Sox2*, *Calm2*, *Smad2*, *Smad5*, *Dnmt3a*, and *Irak1* (G) and inflammatory genes *Il1b*, *Tnf*, *Il6*, *Nos2*, and *Nfkbia* (H) in brain tissue from LPS-injected mice icv injected with vehicle (black) or GW4869 (gray) (*n* = 7).

Data information: Data are displayed as mean ± SEM and analyzed by Student's *t*-test. Significance levels are indicated on the graphs: *0.01 ≤ *P* < 0.05.

---

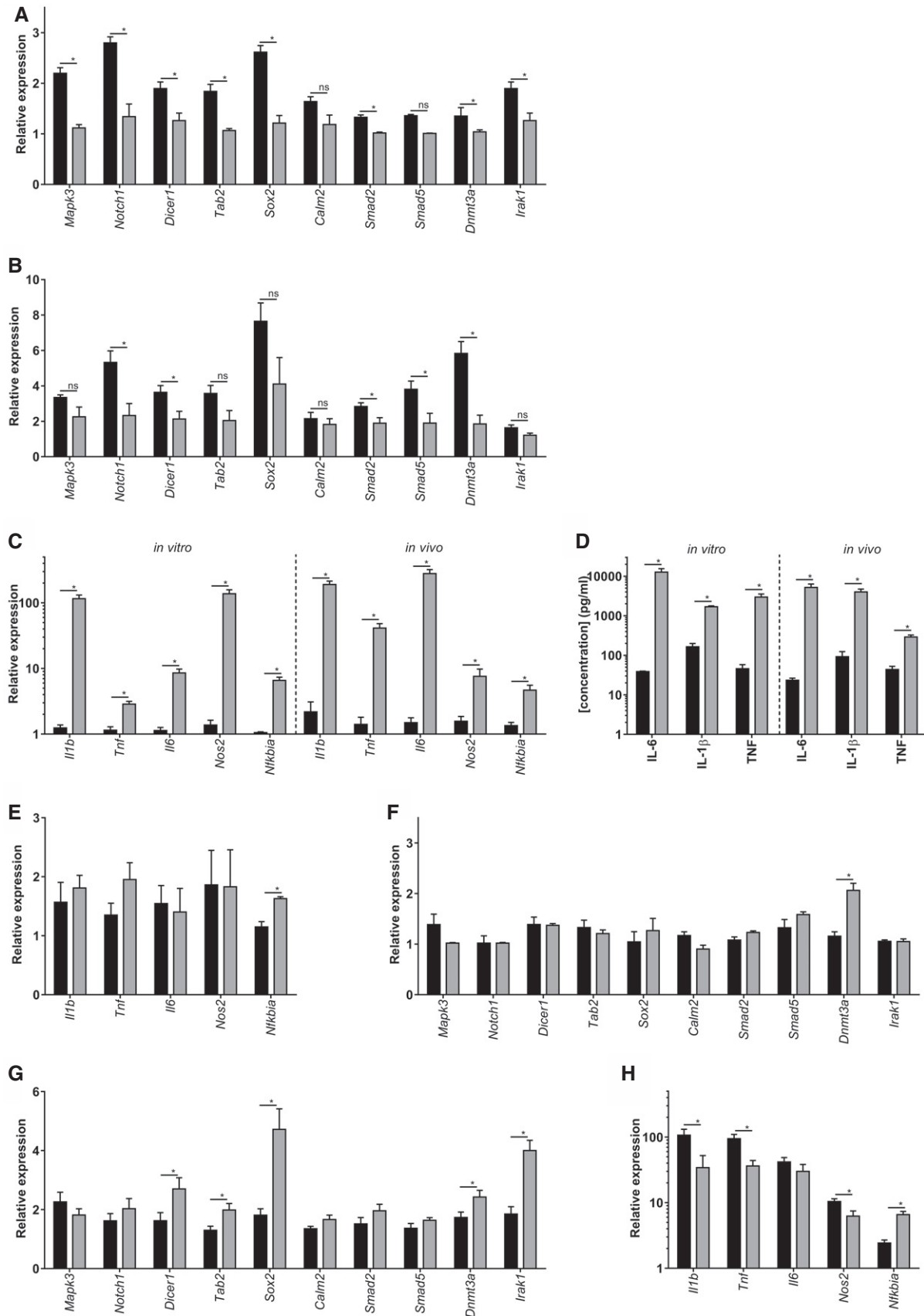

**Figure 8.**

that the observed changes in miRNAs/EVs in the CSF upon LPS stimulation are mainly mediated by CPE cells.

We provide solid evidence that the EVs in the CSF originate from CPE cells. The EVs are positive for the CPE marker TTR, *ex vivo* explant experiments with choroid plexus tissue isolated from control and LPS-injected mice revealed an increase in EV release in the explant supernatant upon LPS injection, *in situ* hybridization of miRNAs that were detected in the CSF revealed expression of the miRs in the CPE cells and several EV proteins showed up-regulation in the choroid plexus upon stimulation with LPS. We also gathered evidence that at least a substantial fraction of the EVs that play an important role in CPE-mediated blood–brain communication are exosomes, membrane-derived vesicles which are secreted from MVBs and mediate intercellular communication by transferring lipids, proteins, mRNAs, and microRNAs to a recipient cell via fusion of the exosome with the target cell membrane (Hannafon & Ding, 2013). Indeed, TEM elegantly showed that MVBs in the CPE cells were completely filled with exosomes in the presence of inflammation, while much less exosomes were present in MVBs of naive mice. Interestingly, the kinetics of the exosomes in the CPE cells resemble the kinetics of the EVs in the CSF. Additionally, systemic inflammation severely affected mRNA and protein levels of genes involved in exosome production in the choroid plexus. An inhibitor of the neutral sphingomyelinase 2 that regulates exosomal miRNA secretion (Trajkovic *et al*, 2008) reduced the induction of EV secretion by LPS in primary CPE cultures and *in vivo* after icv injection in mice, resulting in less miRNA secretion and intracellular miRNA accumulation of several of the miRNAs tested. Moreover, CD63-depletion of the isolated EVs blocked most of the effects of the EVs on the recipient cells. Though these results collectively point to the involvement of exosomes, we do not exclude the possible involvement of microvesicles or lipoprotein-associated miRNAs as well. Moreover, miRNAs associated with other types of RNA-binding proteins, such as AGO2 and NPM1, may also be actively released from donor cells to be taken up by recipient cells, but there is no direct evidence for this (Boon & Vickers, 2013).

Further experiments revealed that the EVs in the CSF can cross the ependymal cells lining the ventricles to reach the brain parenchyma. There, they were taken up by astrocytes and microglia, but not by neurons, indicating that the EVs contain specific proteins that ensure specific targeting and prevent uptake by neurons (Grapp *et al*, 2013). Once taken up, EVs isolated from LPS-injected mice induced down-regulation of miRNA targets and up-regulation of inflammatory genes, causing the recipient cells to produce cytokines and chemokines. Importantly, involvement of LPS contamination in the isolated EVs was excluded by using TLR4$^{-/-}$ primary mixed cortical cultures. Comparison of TLR4$^{-/-}$ and wild-type cortical cultures revealed similar responses to the inflammation-induced EVs. Furthermore, the observed gene expression changes in the recipient cells are clearly exosome dependent, as CD63-depletion of the isolated EVs completely abrogated both the miRNA target repression and the inflammatory gene up-regulation capacity of the isolated EVs. However, the EV effects on the mixed cortical cultures are not exclusively mediated by miRNAs. Indeed, it is known that EVs also contain several other types of molecules, such as mRNA, DNA, and proteins that affect recipient cells. Proteome analysis of the isolated EVs revealed that 280 proteins were specific for the EV proteome after LPS treatment and pathway analysis showed that the inflammatory effect of the EVs is indeed also mediated by proteins.

Here, we focus on the effect of systemic inflammation on the BCSFB, but also the endothelial blood–brain barrier (BBB) is of great importance in the blood-to-brain signaling. Systemic inflammation-induced endothelial cell activation can lead to activation of other cells from the neurovascular unit that are closely connected to the endothelial cells. Consequently, the observed EV production by the choroid plexus upon systemic inflammation will occur in parallel with mechanisms that are activated at the BBB. Therefore, we do not exclude the possibility that the endothelial cells can also produce EVs. However, although some reports studied the role of EVs at the BBB (Haqqani *et al*, 2013; Yamamoto *et al*, 2015; Andras & Toborek, 2016; De Bock *et al*, 2016), it is important to realize that the endothelial cells that form the BBB display minimal vesicle transport activity (Stamatovic *et al*, 2008). In contrast, besides maintaining the barrier, the main function of the CPE cells that form the BCSFB is secretion of proteins into the CSF. Indeed, CPE cells at the BCSFB display much more (vesicular) transport than endothelial BBB cells. Moreover, morphometric studies of the BCSFB revealed that due to the presence of microvilli, CPE cells have an apical surface area in the same size range as the luminal surface area of the BBB endothelial cells, thereby providing an equally large surface for exchange of solutes and vesicles (Keep & Jones, 1990). Unfortunately, probably due to technical reasons, most investigators of the brain barrier totally neglect the choroid plexus. Additionally, we analyzed BBB integrity early upon LPS administration and this revealed that there was no loss of in BBB integrity at this time point, arguing against a role of BBB leakage in the inflammation-induced EV increase in the CSF.

Our data show that inflammatory stimulation of the CPE leads to acute expulsion of EV-associated miRNAs into the CSF that travel to and are taken up by astrocytes and microglia, which in turn respond with an inflammatory program. Indeed, the effects of the peripheral inflammation-induced EV production by the CPE cells could be reversed by the icv injection of an exosome inhibitor and this was reflected by reduced down-regulation and up-regulation of miR target genes and inflammatory genes, respectively.

In most of our experiments, we made use of the endotoxemia model, but we observed increased EV production and similar changes in CPE and CSF miRNA levels in the CLP mouse model of sepsis (Dejager *et al*, 2011), and upon systemic injection of the pro-inflammatory cytokine TNF. Interestingly, both TLR4 and TNFR1 are expressed by the choroid plexus and up-regulated upon systemic inflammation, suggesting that our observations might be the result of a direct response of the choroid plexus to the peripheral inflammatory triggers LPS and TNF. We believe that our study has diagnostic and therapeutic implications. It is known that the CSF is a good indicator of the underlying physiological state of the CNS, and hence, analysis of protein markers in CSF is used for evaluating the clinical state of the CNS (Seehusen, 2010). Our study shows that miRNA analysis of the CSF in pathological conditions could also become useful for diagnosis not only of neurodegenerative diseases (Rao *et al*, 2013), but also of inflammatory conditions, including sepsis, but probably not limited to it. A systematic investigation of CSF miRNA levels in inflammatory models is therefore necessary. Additionally, more researchers are now exploring the therapeutic potential of choroid

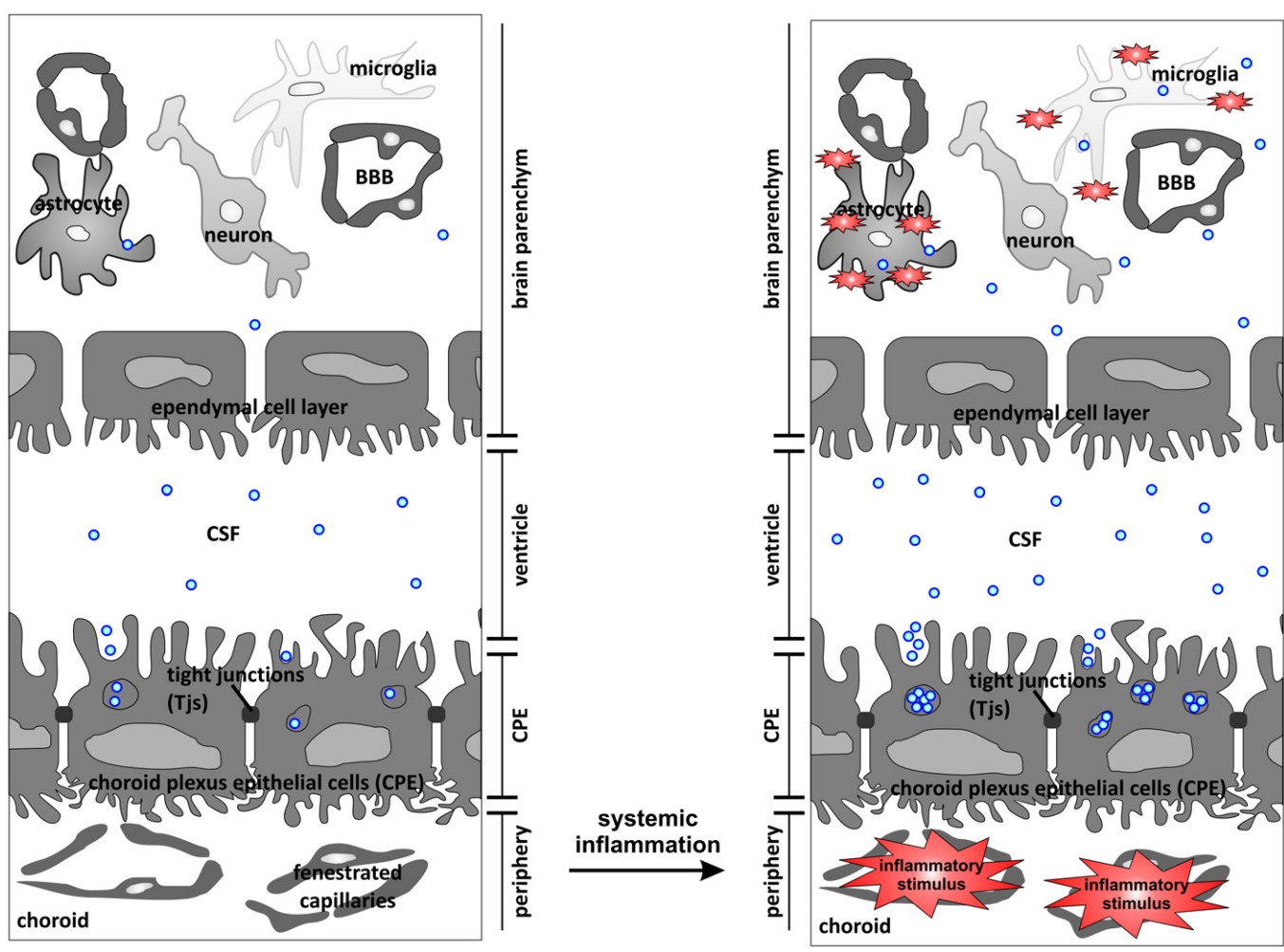

**Figure 9.  Schematic overview of the proposed mechanism.**
Choroid plexus epithelial (CPE) cells form a tight barrier between the peripheral blood and the brain. CPE cells are highly secretory and secrete exosomes from multivesicular bodies (MVBs) into the cerebrospinal fluid (CSF). Systemic inflammation induces activation of the exosome machinery, resulting in increased release of miRNA-containing exosomes into the CSF. Next, these extracellular vesicles (EVs) cross the leaky ependymal cell layer that lines the ventricles and are taken up by astrocytes and microglia, whereupon they transfer a pro-inflammatory message.

plexus as a novel target for immunomodulation to restore brain equilibrium (Baruch & Schwartz, 2013). Although the described choroid plexus epithelial cell transplantation for treatment of Alzheimer's disease (Bolos et al, 2013) is probably not feasible in an acute setting such as sepsis, identification of the involvement of CPE-derived miRNAs in blood–brain communication could lead to the development of new therapeutic strategies, such as icv delivery of anti-inflammatory miRNAs or blockage of CPE miRNA secretion by interfering with EV production. Indeed, although further research is needed, modulation of the miRNA secretion by the CPE could offer protection against neuro-inflammation. Interestingly, Grapp et al (2013) have already shown the delivery of folate into the brain parenchyma by exosomes that are folate receptor-α-positive, suggesting that proteome analysis of the EVs in the CSF could allow cell-specific EVs delivery.

In conclusion, as depicted in Fig 9, we have identified a novel way of blood–brain communication activated by peripheral inflammation via secretion of CPE-derived, miRNA-containing EVs

into the CSF. These EVs carry specific miRNA molecules and proteins, cross the ependymal cell layer lining the ventricles, reach the recipient brain parenchymal cells, trigger target mRNA repression, and induce an inflammatory response. Further research is needed to unravel the mechanisms involved both in the increased EV production by the CPE cells and the EV uptake by the target cells. Moreover, it is tempting to speculate that this process is also important in other inflammatory disorders and could lead to new therapeutic strategies.

# Materials and Methods

### Mice

Female C57BL/6 (8–10 weeks old) mice purchased from Janvier, and TLR4$^{-/-}$ (C57BL/6 background) mice were bred in our facility. Both were housed with 4–6 mice/cage in a specific pathogen-free

animal facility with free access to food and water and with a 14-h light/10-h dark cycle. All experiments were approved by the ethics committee of the Faculty of Sciences of Ghent University.

**Endotoxemia, TNF and CLP model**

Endotoxemia was induced by intraperitoneal (i.p.) injection of lipopolysaccharide (LPS) from *Salmonella enterica* serotype abortus equi (Sigma) dissolved in PBS. The dose was 200 μg/20 g body weight (the $LD_{100}$ dose for C57BL/6 mice). Control animals received i.p. injections of PBS. To study the effect of TNF, mice were injected i.p. with 25 μg/20 g body weight TNF. Severe sepsis was induced in isoflurane-anesthetized mice by ligation of the cecum, followed by twice puncturing with a 21-gauge needle as described earlier (Rittirsch *et al*, 2009). All animals received two doses of antibiotic therapy i.p. (ceftriaxone 25 mg/kg and metronidazole 12.5 mg/kg) 9 and 24 h after CLP. Rectal temperature was measured periodically after challenge.

**CSF isolation**

Cerebrospinal fluid was obtained using the cisterna magna puncture method as described previously (Liu & Duff, 2008). In brief, borosilicate glass capillary tubes (B100-75-15, Sutter Instruments) were used to pull needles on the Sutter P-87 flaming micropipette puller (pressure 330 Pa, heat index 300). Before sampling CSF, mice were sedated with ketamine/xylazine. The incision site was sterilized with 70% ethanol, and cisterna magna was exposed by cutting skin and muscle tissue on the posterior side of the skull. The head of the mouse was placed at an angle of 135°, and CSF was collected by inserting the trimmed needle into the fourth ventricle by piercing the cisterna magna.

**Primary CPE cell cultures**

Primary culture of mouse CPE cells was done as described (Menheniott *et al*, 2010). In brief, pups 2–9 days old were decapitated and the brains were isolated. Choroid plexus from all four ventricles were isolated under a dissection microscope. The cells were dissociated enzymatically by incubating them for 5–7 min with pronase (isolated from *Streptomyces griseus*, Sigma). Digestion was stopped by adding an excess of HBSS buffer, and the cells were washed twice with HBSS. The cell pellet was resuspended in DMEM-F12 culture medium and plated on laminin-coated plates or transwell systems. Two days later, they were shifted to DMEM-F12 containing cytosine arabinoside (Ara-C) to eliminate growth of fibroblasts. Cultures were maintained at 37°C in 5% $CO_2$. Purity of the cultures was confirmed by checking the levels of transthyretin (TTR) both by qPCR and by immunostaining. Primary CPE cells retained the epithelial and the barrier properties which were confirmed by staining for ECDH, ZO1, and occludin and also by measuring the transepithelial electrical resistance (TEER, Millipore). To mimic the *in vivo* situation in the transwell system, LPS stimulation was always done from the basolateral side.

**TEER**

Transepithelial electrical resistance of primary CPE cells was performed using Millicell ERS-2 (MERS00002; Millicell ERS-2

Voltohmmeter; Millipore). Primary CPE cells were trypsinized, and equal number of cells ($10^6$ cells/well) were added to laminin-coated 12-well transwell system (CLS3460-48EA, Sigma). The growth medium was replaced every 48 h. TEER readings were measured every 24 h, according to manufacturer's instructions. In brief, the probe was sterilized by incubating the electrode in 70% ethanol for 15 min and allowed to air dry. Next, the electrode was placed in growth medium for 15 min to equilibrate, followed by TEER measurement of all wells. The blank value (empty wells) was subtracted from the sample readings and multiplied by the effective membrane diameter of the transwell filter. The obtained electrical resistance was expressed in units of $\Omega.cm^2$.

**EV preparation**

Extracellular vesicles were prepared from CSF or cell culture supernatant by using the Total Exosome Isolation kit (Invitrogen) according to the manufacturer's guidelines. For *in vivo* CSF samples, equal volumes of CSF were taken for all conditions and supplemented with 1/20 volume of the Total Exosome Isolation reagent. For isolating EVs from *in vitro* cultures, 2 ml culture supernatant was taken from the transwell system and supplemented with 0.5 volume of Total Exosome Isolation reagent. In both cases, this was followed by overnight incubation at 4°C. Next, samples were spun at 10,000 *g* for 30 min and the pelleted EVs were resuspended in the required volume of PBS. For CD63 depletion of the EVs, the EV pellet was resuspended in 15 μl PBS and 20 μl of the CD63 Dynabeads (Invitrogen) were added to each sample, after washing the beads with wash buffer (0.1% BSA in PBS). This mixture was incubated at 4°C for 18 h and centrifuged at 110 *g* for 15–20 s. Supernatant, that is, the CD63-depleted fraction, was carefully aspirated and used to stimulate the cells. Isolated EVs or CD63-depleted EVs were added to the mixed cortical cultures and incubated for 24 h, followed by supernatant and RNA isolation.

**NanoSight analysis EVs**

For counting the EVs in CSF, debris was removed by centrifugation at 300 *g*, equal volumes of CSF were taken, diluted 1/100 in PBS, and injected into the NanoSight LM10-HS instrument (NanoSight Ltd). Three 60 s videos were recorded for each sample and analyzed with NTA software version 2.3 to determine concentration and size of measured particles with corresponding standard error. NTA post-acquisition settings were optimized and kept constant between samples. Absolute numbers were recorded and back-calculated using the dilution factor. To quantify the EVs *in vitro*, culture supernatant was collected from the transwell system in the presence or absence of the stimulus. Debris was removed by centrifugation at 300 *g*, and the cleared fraction was directly injected into NanoSight without dilution. The NanoSight system was calibrated with polystyrene latex microbeads at 50, 100, and 200 nm (Thermo Scientific, Fremont, USA) prior to analysis.

**Primary mixed cortical cultures**

The cerebral cortex and hippocampus of neonatal wild-type and TLR4$^{-/-}$ mice were obtained after decapitating 3–5 neonatal P1/P2

mice and completely removing the meninges. Tissues were finely minced with a surgical scalpel in cold PBS. Next, samples were spun at 300 $g$ and then digested with trypsin for 15 min at 37°C in a water bath. Cells were again centrifuged at 300 $g$ for 5 min and washed twice with ice-cold PBS. Typically, 3–5 × $10^{5–6}$ cells were obtained from each pup. Cells were resuspended in MEM culture medium (MEM, 10% FBS, Pen/Strep, 2 mM glutamine and non-essential amino acids) and plated directly onto coverslips or six-well plates coated with poly-L-lysine as needed. Cells were maintained in standard tissue culture conditions, and 50% of the medium was replaced once every 2 days. When the cells reached confluency, they were used for visualization of EV uptake or mRNA analysis. The presence of three different cell types was confirmed by staining with GFAP, IBA1 and TUBB3 for astrocytes, microglia, and neurons, respectively. Purified EVs, isolated from the same starting volume of CSF from unchallenged or challenged mice, were incubated on the mixed cortical cultures for 24 h. The amount of EVs added per six-well corresponded to ~12 µl CSF. Next, RNA was isolated from the cell cultures for qPCR analysis and conditioned medium was subjected to cytokine/chemokine analysis.

## Immunoblotting

Choroid plexus samples were lysed in lysis buffer (63 mM Tris–HCl, 2% SDS, 0.1% 2-mercaptoethanol), and protein concentration was determined using the BCA kit (23225ZZ, Life Technologies). Forty micrograms of choroid plexus lysate and EVs isolated from 25 µl of CSF were loaded onto a 15% SDS–PAGE gel. Proteins were separated on the gel at 100 V for 3 h. Proteins were transferred to 0.2 µm nitrocellulose membranes (NBA083G, Perkin Elmer), and membranes were blocked using Odyssey blocking buffer (927-40000, Li-Cor). Next, membranes were incubated with TTR (1:1,000; A002, DAKO) and β-actin (1:5,000; 691002, MP Biomedicals) antibodies diluted in Odyssey blocking buffer. After washing, blots were incubated with IRDye 800CW goat anti-rabbit IgG (1:10,000; LI 926-32211, Li-Cor) or IRDye 680 goat anti-mouse IgG (1:10,000; LI 926-68070, Li-Cor) and imaging was done using the Odyssey imaging system.

## Immunohistochemistry

Primary cells were fixed with 4% PFA for 30 min on ice or with methanol for RiboGreen staining. Fixed cells were washed three times with PBS and permeabilized with 0.1% Triton X-100 for 30 min on ice. Samples were blocked with 5% BSA for 1 h at room temperature and incubated with primary and secondary antibodies (diluted in 2% BSA) overnight and for 1 h at room temperature, respectively. The following antibodies were used: ZO1 (2 µg/ml; 617300, Invitrogen), ECDH (1:100; 610181, BD), CLDN1 (5 µg/ml; 51-9000, Invitrogen), CD63 (1:200; sc-31214, Santa Cruz Biotechnology), IBA1 (1 µg/ml; 019-19741, Wako), GFAP (1:10,000; Z0334, DAKO), and beta III tubulin (TUBB3; 1:100; AB9354, Millipore). As secondary antibodies, goat anti-rabbit-DyLight 633 (1:400) and goat anti-rabbit-DyLight 594 (1:400) were used. Slides were counterstained with Hoechst (1 µg/ml) and mounted using 2% n-propyl gallate. For RNA staining with RiboGreen (Ganguly et al, 2009; Chiba et al, 2012), CPE cells were cultured on laminin-coated coverslips until reaching confluency and treated with LPS. After 6 h of treatment, cells were fixed with ice-cold methanol for 15 min, washed with PBS for three times, and incubated with RiboGreen (1:200, Invitrogen) for 30 min at room temperature. Next, samples were counterstained with Hoechst (1 µg/ml) for 20 min at room temperature, washed and mounted using 2% n-propyl gallate. For immunofluorescent analysis of EVs, CSF was diluted with PBS and incubated with PKH26 (1:200, Sigma) and RiboGreen (1:200, Invitrogen) for 20 min at room temperature. This was followed by exosome isolation and pellet resuspension in PBS. For immunostainings on brain sections, mice were transcardially perfused with PBS and brains were isolated. For CD63, RAB5, and ANXA2, brains were incubated overnight in 4% PFA, followed by paraffin embedding and sectioning. Sections were treated with citrate buffer (S2031, DAKO) and incubated overnight with the following primary antibodies: CD63 (1:200; sc-31214, Santa Cruz Biotechnology), RAB5 (1:500; ab18211, Abcam), and ANXA2 (1:200; ab54771, Abcam). After washing, secondary antibodies were added [goat anti-rabbit biotin (1:500; E0432, DAKO) and goat anti-mouse biotin (1:500; E0433, DAKO)], followed by an amplification step using the ABC system (PK-6100, Vector laboratories) and TSA (SAT700001EA, Perkin Elmer) according to manufacturer's instructions. For TLR4 and TNFR1, brains were after isolation immediately immersed in cryoprotectants stored at −80°C and 30 µm sections were made using the HM 560 CryoStar (Thermo Scientific™). Sections were air-dried, followed by 30-min fixation with 1% PFA and blocking with 2% BSA. Next, sections were incubated with TLR4 (1:200; 13-9924-81, eBiosciences), TNFR1 (1:200; AF-425-SP, R&D Systems), and pan-cytokeratin (1:500; ab7753, Abcam) antibodies overnight. After a washing step, secondary antibodies were added: anti-streptavidin-DyLight 633, donkey anti-goat-DyLight 633, and goat anti-mouse-DyLight 488. Next, samples were counterstained with Hoechst (1 µg/ml) and the sections were mounted. A Leica TCS SP5 II confocal microscope was used for visualization.

## TEM

To visualize EVs by TEM, undiluted CSF samples were adsorbed for 10 min on a formvar-/carbon-coated grid. Samples were negatively stained with 2% uranyl acetate. To visualize the CPE cells, isolated choroid plexus was fixed in a solution of 0.3% glutaraldehyde and 2.5% formaldehyde dissolved in 0.1 M sodium cacodylate buffer containing 20 mg/100 ml $CaCl_2$. Fixed specimens were dehydrated through a graded ethanol series and embedded in Spurr's resin. Ultrathin sections of a gold interference color were cut using an ultramicrotome (Leica EM UC6) and post-stained with uranyl acetate for 40 min and lead citrate for 7 min in a Leica EM AC20. Sections were collected on formvar-coated copper slot grids. Samples were examined with a transmission electron microscope (JEOL 1010; JEOL, Tokyo, Japan).

## In vivo brain injections

For injection of EVs into the lateral cerebral ventricles (icv) of the mice, mice were anesthetized with isoflurane and placed on a stereotactic frame. Body temperature was maintained at 37°C using a heating pad. The position of the lateral ventricle was calculated using coordinates of bregma and lambda. Then, 4 µl of EVs labeled with PKH26 (corresponding with ~2.9 × $10^8$ particles) was injected

into the ventricles, and after 4 h, mice were sacrificed and brain samples were isolated. Brain samples were immediately placed in cryomolds with cryoprotectant (Thermo Scientific™) and stored at −80°C. Cryosections (50 μm) were made from the frozen brains using HM 560 CryoStar (Thermo Scientific™), sections were fixed and stained immediately, pan-cytokeratin (1:500; ab7753, Abcam) was used as a marker for choroid plexus epithelium, and Hoechst (1 μg/ml) was used to stain the nucleus. Chemical inhibition of exosome production *in vivo* was performed by icv injection of GW4869, a neutral sphingomyelinases inhibitor (nSMase2). Two microliters of GW4869 (4.3 mM in DMSO) or vehicle was injected into the lateral ventricles of the mice 4 h after i.p. injection of LPS (200 μg/20 g body weight). CSF and choroid plexus were isolated 2 h after GW4869 administration for EV and miRNA quantification.

### Cytokine/chemokine measurements

Cytokines and chemokines in CSF and cell culture supernatants were measured by the Bio-Plex cytokine assays (Bio-Rad) according to the manufacturer's instructions.

### Analysis of BBB and BCSFB leakage

Blood–brain barrier and blood-CSF barrier leakage was analyzed as described previously (Vandenbroucke *et al*, 2012). In brief, FITC-dextran was injected intravenously (i.v.) 3 h after LPS, followed by CSF isolation by cisterna magna puncture 1 h later. For BCSFB analysis, 1 μl of CSF was diluted in 99 μl PBS and fluorescence was measured using a fluorometer ($\lambda_{ex}/\lambda_{em}$ = 488 nm/520 nm). For the analysis of BBB integrity, mice were transcardially perfused with PBS and brain was isolated. Next, perfused brain samples were incubated in formamide overnight, samples were centrifuged at high speed, and 100 μl supernatant was used for fluorescence measurement using a fluorometer ($\lambda_{ex}/\lambda_{em}$ = 488 nm/520 nm).

### RNA isolation and quality control

Total RNA was isolated with the miRNeasy kit (Qiagen). For isolation of total RNA from *in vivo* choroid plexus samples, mice were anesthetized with ketamine/xylazine and perfused with PBS supplemented with bromophenol blue. Brains were dissected from the skull, and choroid plexus tissue from the third and fourth ventricles was isolated. Total RNA was isolated without pooling samples. For isolating total RNA from *in vivo* CSF samples, equal volumes of pooled CSF samples (total volume 50 μl) were taken from all conditions and used for total RNA isolation. For RNA isolation from primary CPE cells, confluent cultures grown on transwells were treated with LPS from the basal side for various durations, cells were lysed, and RNA was isolated. For RNA isolation from culture supernatant, equal amounts of culture supernatant were taken from the transwell system, EVs were precipitated using the Total Exosome Isolation reagent (Invitrogen), and total RNA was isolated from the exosomal pellet. For isolating RNA from the mixed cortical culture, purified EVs were added to the cells and after 24 h, and total cellular RNA was isolated. RNA concentration and purity were determined spectrophotometrically using the Nanodrop ND-1000 (Nanodrop Technologies), and RNA integrity was assessed using a Bioanalyzer 2100 (Agilent).

### qPCR mRNA and miRNA analyses

For mRNA qPCR, cDNA was made by using iScript™ cDNA Synthesis Kit (Bio-Rad) with 500–1,000 ng starting material. qPCR was done using the SensiFAST™ SYBR No-ROX kit (Bioline) on the Light Cycler 480 system (Roche). Expression levels were normalized to the expression of the two most stable housekeeping genes, which were determined for each experiment using the geNorm Housekeeping Gene (HKG) Selection Software (Vandesompele *et al*, 2002). All primer sequences can be found in Appendix Table S1. The HKGs for *in vivo* choroid plexus samples were UBC and GAPDH for primary CPE cells, RPL and GAPDH for mixed cortical cultures, and UBC and RPL for *in vivo* brain tissue. For miRNA qPCR, cDNA was prepared using MultiScribe™ Reverse Transcriptase (Invitrogen™) kit, starting from equal amounts of total RNA (350 ng) for choroid plexus and equal volumes (4 μl) for CSF. Next, qPCR was done using TaqMan qPCR assays according to the manufacturer's instructions. Selection of stable miRNAs for normalization was done based on available miRNA profiling data using the geNorm Housekeeping Gene (HKG) Selection Software. miR-24 and miR-126 were used as HKG miRNAs for both *in vitro* and *in vivo* choroid plexus samples. For miRNA normalization in CSF and culture supernatant, expression normalization was done by using overall expression.

### Choroid plexus explants

Mice were injected i.p. with PBS or LPS (200 μg/20 g body weight) and two and a half hours later transcardially perfused with serum-free DMEM. Choroid plexus tissue was isolated from all four ventricles and incubated in 200 μl OptiMEM for two and a half hours at 37°C in 5% $CO_2$. Conditioned medium was collected, centrifuged at 300 *g* for 5 min to remove cellular debris, and supernatant was analyzed by NanoSight.

### *In situ* hybridization

Mice were injected i.p. with PBS or LPS (200 μg/20 g body weight) and transcardially perfused with 4% PFA 6 h later. Next, brains were isolated and embedded in paraffin. Sections of 5 μm thick were made and mounted onto SuperFrost plus glass slides. ISH was performed using miRCURY LNA microRNA ISH optimization kit (Exiqon, 90010) according to manufacturer's instructions. In brief, sections were deparaffinized, digested with proteinase K for 10 min at 37°C, and washed with PBS. Next, slides were dehydrated and incubated with 25 μM of double-DIG LNA™ microRNA probe (miR-155, miR-146a, and miR-9) and hybridization was performed overnight at 56°C in a humidifying chamber. Post-hybridization washes were done with SSC solution (5× SSC for 5 min, 1× SSC for 5 min and 0.2× SSC for three times at hybridization temperature). The slides were blocked with 2% sheep serum supplemented with 1% BSA in PBS-T (PBS/0.05% Tween-20) for 30 min at room temperature. After washing with PBS, anti-DIG reagent (Roche, sheep anti-DIG-AP at 1:800) was applied onto the slides and incubated at room temperature for one hour. Then, slides were washed with PBS-T for three min, three times. Next, freshly prepared AP substrate (Roche, NBT/BCIP ready use tablets; 11697471001) was applied onto sections and

incubated for 1 h at 30°C in a humidified dark chamber. KTBT solution (50 mM Tris–HCl, 150 mM NaCl, and 10 mM KCl) was applied on the slides to stop the reaction, and the slides were washed three times 3 min with PBS and mounted using 2% n-propyl gallate.

## Proteome analysis of EVs and CPE cells

Extracellular vesicles were isolated from 50 μl of CSF using Total Exosome Isolation reagent (4478359, Life Technologies) and subsequently lysed in 25 μl of SDS–PAGE loading buffer containing XT Reducing Agent (Bio-Rad). Each sample was then loaded onto a 4–12% precast gradient gel (Criterion XT, Bio-Rad) and separated until the solvent front penetrated about 1 cm into the separation gel. The proteins were stained using SimplyBlue™ SafeStain (Invitrogen). Per EV sample, a single stained protein band was excised and washed with 100 μl of water, 50% and 100% acetonitrile for 15 min each. The washed gel bands were then completely dried under vacuum. In-gel digestion was performed using trypsin (Promega; V5280; Trypsin Gold, Mass Spectrometry grade) at a final concentration of 13 ng/μl and in a total 200 μl of 50 mM ammonium bicarbonate buffer (pH 8). Digestion was allowed to take place overnight at 37°C.

The generated peptide mixture was centrifuged to remove possible debris, and the peptides were collected, vacuum dried, and re-dissolved in 80 μl of 2% acetonitrile and 0.1% TFA prior to pre-fractionation by RP-HPLC [C18-HD; 3 μm beads, 12 cm column with 250 μm internal diameter (I.D.)]. Initially, 10 min of isocratic pumping of solvent A [10 mM ammonium acetate in water/acetonitrile (98/2, v/v) at pH 5.5] was applied. Then, the peptides were separated using linear gradient from 100% solvent A to 100% solvent B (10 mM ammonium acetate in acetonitrile/water (70/30, v/v), at pH 5.5) for 30 min at a flow rate of 3 μl/min. Peptides eluting between 0 and 80 min were collected at a time interval of 1 min each, further pooled into 20 fractions, vacuum dried, and re-dissolved in 12 μl of 2% acetonitrile and 0.1% TFA. Of each fraction, 6 μl was introduced into an LC-MS/MS system; an Ultimate 3000 RSLC nano LC (Thermo Fisher Scientific) in-line connected to a Q-Exactive mass spectrometer (Thermo Fisher Scientific). Peptides were first trapped on a trapping column [made in-house, 100 μm I.D. × 20 mm, 5 μm C18 Reprosil-HD beads (Dr. Maisch, Ammerbuch-Entringen, Germany)]. Subsequently, peptides were loaded on to analytical column [made in-house, 75 μm I.D. × 150 mm, 3 μm C18 Reprosil-HD beads (Dr. Maisch)] which was packed in the nanospray needle (PicoFrit SELF/P PicoTip emitter, PF360-75-15-N-5, New Objective, Woburn, MA, USA). Peptides were initially loaded using 0.1% TFA and separated using a linear gradient from 98% of solvent A' (0.1% formic acid) to 55% of solvent B' [0.1% formic acid in water/acetonitrile, 20/80 (v/v)] for 30 min at a flow rate of 300 nl/min. This was followed by a 5-min wash reaching 99% of solvent B'. The mass spectrometer was operated in data-dependent, positive ionization mode, automatically switching between MS and MS/MS acquisition for the 10 most abundant peaks in a given MS spectrum.

For preparation of choroid plexus cell lysates, choroid plexus was collected and pooled from five mice, lysed by mechanical disruption (three cycles at 20 Hz speed for 30 s each, Retsch) using an metal bead in Laemmli sample buffer (2% SDS, 10% glycerol, 120 mM Tris–HCl pH 6.8, 0.02% bromophenol blue). After lysis, cell lysates were centrifuged at high speed to remove any debris and supernatants were collected, separated by SDS–PAGE and in-gel digested as described above for the EV samples. Peptide mixtures were labeled with freshly prepared N-hydroxy-succinimide esters of different forms of propionate; control samples were labeled with light isotopes ($^{12}C_3$-propionate) and LPS-treated samples with heavy isotopes ($^{13}C_3$-propionate) as previously described (Ghesquiere *et al*, 2011). Equal amounts of peptide samples were mixed and analyzed by LC-MS/MS as described above.

## Proteome data analysis

The mass spectrometry proteomics data have been deposited in the ProteomeXchange Consortium via the PRIDE partner repository (dataset identifier PXD001346 and DOI: 10.6019/PXD001346; dataset identifier PXD001347 and DOI: 10.6019/PXD001347). Mascot Daemon (version 2.4, Matrix Science) was used to identify the generated MS/MS spectra. Here, acetylation of protein N-termini, oxidation of Met, propionamide modification of Cys, and pyroglutamate formation (N-terminal Gln) were set as variable modifications. Further, trypsin was set as the used protease with one missed cleavages allowed. The peptide mass tolerance was set at 10 ppm, and the mass tolerance on fragment ions was set to 20 millimass units (mmu). The machine setting was set to ESI-QUAD, and spectra were searched against the mouse section of the Swiss-Prot database. Only peptide-to-spectrum matches with the highest score above the corresponding threshold score when using a confidence level of 95% and with at least seven amino acids were considered for further analysis. For label-free quantification, MaxLFQ with similar search parameters as for the Mascot search. The LFQ intensity values obtained by MaxLFQ were used for data analysis in the Perseus environment. Here, the intensity values were logarithmized, and proteins with at least two peptides were withheld for further quantification. Missing values were replaced with the help of the imputation function of Perseus based on a normal distribution of LFQ intensities (width 0.3 and standard deviation of 1.8). Multiple scatter plots with Pearson correlation were used to assess for correlation between the samples within a given group and between groups. Protein, peptide, and site false discovery rates (FDR) were adjusted to 0.08. All proteins identified with two or more peptides were taken for label-free quantification analysis. Ingenuity Pathway Analysis (Ingenuity Systems Inc., Redwood City, CA, USA) was used for identifying enriched pathways. The DAVID knowledgebase was used for GO term enrichment analysis, and here, the choroid plexus proteome was used as the background list for the EV proteome that was used as the query list.

## Statistics

Data are presented as means ± standard error of mean (SEM). Data were analyzed by Student's *t*-test. Significance levels are indicated on the graphs: $*0.01 \leq P < 0.05$; $**0.001 \leq P < 0.01$; $***0.0001 \leq P < 0.001$; $****P < 0.0001$. Exact *n*- and *P*-values of the significant results can be found in Appendix Table S2.

## The paper explained

### Problem

Systemic inflammatory response syndrome (SIRS) is a systemic inflammatory disease caused by, for example, trauma, burns, and infection. No treatment is available except support of vital functions and antibiotics in case of an infection. Therefore, understanding the pathology of SIRS is of extreme importance for finding new therapeutic targets.

### Results

Here, we provide evidence that the choroid plexus epithelial (CPE) cells, which are strategically positioned at the interface of blood and cerebrospinal fluid (CSF, the fluid that fills the ventricles in the brain), play a central role in the communication between blood and brain. We identified a novel way of blood-to-brain communication which is activated by peripheral inflammation and occurs via secretion of CPE-derived extracellular vesicles (EVs) into the CSF. These CPE-derived EVs carry specific miRNA molecules and proteins, cross the ependymal cell layer lining the ventricles, reach specific brain parenchymal cells, and induce both mRNA repression and inflammatory gene activation.

### Impact

The identification of this novel mechanism of blood-to-brain communication might open up new strategies to treat systemic inflammation. Moreover, further research is needed to determine whether this is a universal inflammation-activated mechanism that plays a role in different neurological and/or systemic diseases.

**Expanded View** for this article is available online.

## Acknowledgements

This work was supported by the Agency for Innovation of Science and Technology in Flanders, the Research Council of Ghent University, the Research Foundation Flanders (FWO Vlaanderen), the Foundation for Alzheimer Research (SAO-FRA), COST action BM1402, and the Interuniversity Attraction Poles Program of the Belgian Science Policy (IAP-VI-18). Authors wish to thank the VIB Nucleomics Core for the high-throughput analysis, Hans Demol and Evy Timmerman for technical assistance with the proteome analysis, Joke Vanden Berghe for technical assistance related to the breeding the mice, Dr. Amin Bredan for editing the manuscript, and the VIB Bio Imaging Core (especially Dr. C. Guérin and Dr. S. Lippens) for suggestions and helpful discussions concerning the microscopy studies.

## Author contributions

SB performed most of the experiments with assistance of EVW, REV, MB, and DD, KG helped with the proteome data analysis, RDR performed the EM imaging, and AH assisted in the NanoSight analysis. KR and SS gave valuable input related the extracellular vesicle analysis. VV helped with the mixed cortical cultures. REV and CL supervised the experiments and wrote the manuscript with assistance of SB.

## Conflict of interest

The authors declare that they have no conflict of interest.

## For more information

The mass spectrometry proteomics data have been deposited in the ProteomeXchange Consortium via the PRIDE partner repository (dataset identifier PXD001346 and DOI: 10.6019/PXD001346; dataset identifier PXD001347 and DOI: 10.6019/PXD001347).

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
