## [Review Process File · EMBO Molecular Medicine]

Identification of a novel mechanism of blood–brain communication during peripheral inflammation via choroid plexus-derived extracellular vesicles

Sriram Balusu, Elien Van Wonterghem, Riet De Rycke, Koen Raemdonck, Stephan Stremersch, Kris Gevaert, Marjana Brkic, Delphine Demeestere, Valerie Vanhooren, An Hendrix, Claude Libert and Roosmarijn E. Vandenbroucke

Corresponding author: Roosmarijn Vandenbroucke, Ghent University/VIB

Review timeline:

Submission date:	03 February 2016
Editorial Decision:	26 February 2016
Revision received:	04 June 2016
Editorial Decision:	15 June 2016
Revision received:	11 July 2016
Accepted:	27 July 2016

Transaction Report:

Editor: Céline Carret

1st Editorial Decision

26 February 2016

Thank you for the submission of your manuscript to EMBO Molecular Medicine. We have now heard back from the three referees whom we asked to evaluate your manuscript. Although the referees find the study to be of potential interest, they also raise a number of concerns that need to be addressed in the next final version of your article.

You will see from the comments below, that while referee 1 is rather supportive, referees 2 and 3 raise several issues that weaken the findings. Nevertheless, constructive suggestions and comments are provided that we feel, if followed thoroughly, would considerably improve the conclusiveness of the data and as such, I would strongly encourage you to address all issues raised as recommended.

Given these evaluations, I would like to give you the opportunity to revise your manuscript, with the understanding that the referees' concerns must be fully addressed and that acceptance of the manuscript would entail a second round of review. Please note that it is EMBO Molecular Medicine policy to allow only a single round of revision and that, as acceptance or rejection of the manuscript will depend on another round of review, your responses should be as complete as possible.

Please read below for important editorial formatting.

I look forward to receiving your revised manuscript.

***** Reviewer's comments *****

Referee #1 (Comments on Novelty/Model System):

This an investigation of a great quality offering a brand new explanation on the induction of brain inflammation after systemic insults (LPS, peritonitis, TNF). With elegant in vitro and in vivo experiments the authors provide most convincing data.

Referee #1 (Remarks):

Balusu and colleagues have identified how microvesicles derived from choroid plexus, which contain miRNA can propagate the inflammatory response within the brain. This constitutes a new key contribution provided by the members of the Inflammation Research Center of Ghent. The work is elegantly carried out with both in vitro and in vivo experiments, and all additional experiments a reviewer could think about have already been achieved!

Minor comments

- 1/ p.5. The authors wrote : "miRNAs are an important cargo of EVs". Would it not be the reverse?
- 2/ There are some discrepancies between the results of miRNAs found in microvesicles and within the cells. Did the authors checked at different timing post-LPS to confirm their proposed hypothesis to explain these observations?
- 3/ p.9 & p.18. The authors refer to their own previous review (ref. 45) to claim that CLP is the best model of human sepsis. This is a bit exaggerated as compared to what is stated in this review: "This demonstrates that experimental sepsis models do not completely mimic human sepsis." Let's say it is a model widely used with a lot of limitation. The main one being that mice are far to mimic humans!
- 4/ Please recall in fig.5 legend the amount of injected TNF.
- 5/ What is the mortality in the used CLP model?
- 6/ The authors focus on the induction of pro-inflammatory cytokines. Could EVs as well and simultaneously induce anti-inflammatory ones (IL-10, IL-1ra, TGFb...)?

Referee #2 (Remarks):

In this study, Balusu and colleagues examined the response of the choroid plexus to LPS stimulation in-vivo and in-vitro. They show elevated levels of exosomes in the cerebrospinal fluid and link this to CP production of exosomes. They further suggest that pro-inflammatory miRNA which are carried inside those exosomes, reach and affect the brain parenchyma.

The manuscript is well written and the figures are generally clear. Though the topic of the manuscript is interesting, the novelty is limited taking into account the previous reports which A) characterized in vitro and in vivo the response of the choroid plexus to LPS and other pro-inflammatory mediators, B) characterized CSF exomes and their content following inflammatory stimuli, and C) characterized the effect of exosomes and pro-inflammatory miRNA on neural tissues. Therefore, it seems that the novelty of the current work should be primarily judged on the question of whether there is any functional contribution of choroid plexus-secreted exosomes in the LPS model, in comparison to simply BBB leakiness to the CSF.

Major concerns:

1. Figure 1 data shows that very shortly (1-2 hours) after mice get i.p. LPS injection, there is elevation of exosomes and pro-inflammatory miRNAs in the CSF. Figure 4, which suggests that the CPE is the major source of these exosomes, shows that 3-4 hours after LPS stimulation there is elevation in the CPE of genes related to exosome secretion. If the CPE was the main source of exosomes found in the CSF, these should have appeared in the hours after the CPE starts to produce them. Therefore, it seems that the data simply suggests that there is an influx of pro-inflammatory mediators from the plasma to the CSF following LPS injection. This phenomenon is widely documented in the literature in the context of BBB leakiness.

It is clear that the CPE can also produce exosomes, as was previously reported, and as the authors show in the *in vitro* model of figure 2, but the relative contribution of this in comparison to BBB leakiness was not addressed in the current manuscript, and is a major weakness of the suggested hypothesis. If the authors can address this question experimentally it would greatly improve the manuscript.

2. The authors show data of several experiments in which they use the compound GW4869, which they refer to as "exosome inhibitor", to demonstrate that the pro-inflammatory effect and genes expression observed are exosome-dependent. Since GW4869 is a classical NF κ B and TNF- α blocker, the observed effects are very much expected and do not contribute to the hypothesis presented. For example, in the data presented in figure 3, is it surprising that LPS-induced pro-inflammatory genes expression is downregulated following exposure to an NF κ B-blocker? This output repeats itself in various experiments along the manuscript, and should be cautiously regarded in terms of any mechanistic-insight to the exosomes.

3. Figure 6 shows massive amounts of data from protein analysis. Most of the data is not statistically significant (and is particularly accustomed to be presented in Z score and not P value, as per the multiple testing provided). Moreover, it is not appropriate to selectively choose from the GO enrichment terms specific biological processes of interest which fit a narrative (page 11), but to provide their significance. A short list of the top 5/10/X statistically significant changes would be much more informative to the reader.

4. The data presented in figure 7 is important to the hypothesis presented in the manuscript, as it is basically the only link between the CSF exosomes, which have to go pass the ependymal layer, and the effect on the brain parenchyma. Unfortunately, the presented pictures are not convincing. Figure 7a,b show DAPI positive cells in the CSF, which are for some reason in proximity to GFAP positive astrocytes - where anatomically were these pictures taken?

It is not clear at all from the picture how come the "CP", the "CSF", and the "brain" labeling are positioned where they are. There are specific immunohistochemistry markers which stain the ependymal layer of the brain ventricles; these should be used to show that the exosomes pass the ependymal layer. Also, to label the choroid plexus tissue, a specific marker should be used, such as TTR, or epithelial markers.

Minor comments:

5. In the legends to several figures the n values provided should be examined again. For example, figure 2c, 2e, and 3b, graphs have bars without standard errors, though in the legends it is described as n=3 or n=4. Were samples omitted from the groups? If so, please provide the specific n value for each group after samples were omitted. In several cases it looks like n=1 after samples were omitted.

6. In the legends to the figures and the discussion there are several instances where *in vitro* experiments with LPS treatments are described as "systemic inflammation". It would be clearer if clearly stated as "LPS treatment".

Referee #3 (Remarks):

Balusu et al. present a very interesting and original manuscript on the role of choroid plexus in the communication between the blood and the brain during peripheral inflammation, suggesting that the process is mediated by extracellular vesicles, mostly exosomes.

In addition to its originality and significance, the work shows a detailed and thorough description and characterisation of of this phenomenon.

Some reservations relate to the recurring claim about the sensing capacity of the choroid plexus as no direct evidence is presented, only correlative data. At the very least, the authors should present evidence for the presence of receptors that can sense inflammation triggers, according to the model used. For example, the authors can try to use IHC to detect TLR4 in the case of LPS, or TNFR in the case of a TNF-driven model of inflammation. An alternative scenario is for example that either LPS or TNF alter the permeability of the BBB that then allows the same mediators to be sensed directly by microglia.

Another aspect that deserves clarification is the implication that up regulation of mir146a contributes to inflammation as the literature largely favours the opposite effect.

Finally, it would be interesting to investigate whether brain markers of inflammation are decreased in animals with genetic deficiencies in factors involved in exosome secretion such as Rab27.

1st Revision - authors' response

04 June 2016

Referee #1

This is an investigation of a great quality offering a brand new explanation on the induction of brain inflammation after systemic insults (LPS, peritonitis, TNF). With elegant *in vitro* and *in vivo* experiments the authors provide most convincing data. Balusu and colleagues have identified how microvesicles derived from choroid plexus, which contain miRNA can propagate the inflammatory response within the brain. This constitutes a new key contribution provided by the members of the Inflammation Research Center of Ghent. The work is elegantly carried out with both *in vitro* and *in vivo* experiments, and all additional experiments a reviewer could think about have already been achieved!

We thank the referee for these positive comments!

Minor comments

p.5. The authors wrote: "miRNAs are an important cargo of EVs". Would it not be the reverse?

This has been corrected in the revised manuscript.

There are some discrepancies between the results of miRNAs found in microvesicles and within the cells. Did the authors checked at different timing post-LPS to confirm their proposed hypothesis to explain these observations?

We appreciate this critical comment and discussed this more in detail in the revised manuscript. We analyzed miRNA levels at 1 and 6 h post LPS in the CSF (Figure 1d-g in our manuscript) and 1, 6 and 24 h after LPS in the choroid plexus (Figure 4i-l in our manuscript). This revealed upregulation of miR-1a, miR-9, miR-146a and miR-155 in CSF upon systemic inflammation. In the choroid plexus however, only two miRNAs displayed upregulation (miR-146a and miR-155), while the two other miRNAs (miR-1a and miR-9) were downregulated at the same time points. This difference

might be explained by differences in the ratio of transcriptional activation versus secretion. This means that the secretion of miR-1a and miR-9 exceeds their transcriptional upregulation, eventually resulting in a downregulation in the choroid plexus; in contrast to miR-146a and miR-155. Although determining the mechanism of inflammation-induced miRNA loading into the vesicles and analysis of the role of the specific miRNAs is for sure interesting to be studied in the future, we don't think that analyzing additional time points will have added value to our current manuscript.

p.9 & p.18. The authors refer to their own previous review (ref. 45) to claim that CLP is the best model of human sepsis. This is a bit exaggerated as compared to what is stated in this review: "This demonstrates that experimental sepsis models do not completely mimic human sepsis." Let's say it is a model widely used with a lot of limitations. The main one being that mice are far to mimic humans!

We have revised the sentence according to the reviewer's suggestion.

Please recall in fig. 5 legend the amount of injected TNF.

We have used TNF at a concentration of 25 μ g/20 g body weight as mentioned in the materials and methods and this is now also incorporated in the figure legend of the revised manuscript.

What is the mortality in the used CLP model?

The mortality of the used CLP model is ~75%. We included this information in the revised manuscript.

The authors focus on the induction of pro-inflammatory cytokines. Could EVs as well and simultaneously induce anti-inflammatory ones (IL-10, IL-1ra, TGFb...)?

This is an interesting comment. To address this, we checked the presence of anti-inflammatory molecules such as IL-10, IL-1RA, and TGF β in our EV proteomics list. Although this list might not be exhaustive, we did not identify these proteins in the EVs. Additionally, we analyzed whether IL10 was present in the supernatant of mixed cortical cultures incubated with EVs. As shown in **Figure 1**, IL10 protein was increased in the supernatant of the cells incubated with EVs derived from LPS injected mice, showing that EVs also induce the anti-inflammatory protein IL10.

Figure 1. Analysis of the pro-inflammatory effect of EVs isolated from LPS-injected mice on mixed cortical cultures. Bio-Plex cytokine assay (Bio-Rad) of IL10 in the supernatant of mixed cortical cultures after incubation for 24 h with EVs isolated from CSF from untreated (black) or LPS treated (6 h) (grey) mice (n=3).

Referee #2

In this study, Balusu and colleagues examined the response of the choroid plexus to LPS stimulation *in-vivo* and *in-vitro*. They show elevated levels of exosomes in the cerebrospinal fluid and link this

to CP production of exosomes. They further suggest that pro-inflammatory miRNA which are carried inside those exosomes, reach and affect the brain parenchyma.

The manuscript is well written and the figures are generally clear. Though the topic of the manuscript is interesting, the novelty is limited taking into account the previous reports which A) characterized *in vitro* and *in vivo* the response of the choroid plexus to LPS and other pro-inflammatory mediators, B) characterized CSF exosomes and their content following inflammatory stimuli, and C) characterized the effect of exosomes and pro-inflammatory miRNA on neural tissues. Therefore, it seems that the novelty of the current work should be primarily judged on the question of whether there is any functional contribution of choroid plexus-secreted exosomes in the LPS model, in comparison to simply BBB leakiness to the CSF.

The novelty of our manuscript is indeed that we are the first to show that the choroid plexus epithelial cells secrete more EVs into the CSF upon systemic inflammation and that these EVs transfer a pro-inflammatory signal to the brain, in parallel with what is happening at the BBB.

Major concerns:

Figure 1 data shows that very shortly (1-2 hours) after mice get i.p. LPS injection, there is elevation of exosomes and pro-inflammatory miRNAs in the CSF. Figure 4, which suggests that the CPE is the major source of these exosomes, shows that 3-4 hours after LPS stimulation there is elevation in the CPE of genes related to exosome secretion. **If the CPE was the main source of exosomes found in the CSF, these should have appeared in the hours after the CPE starts to produce them.** Therefore, it seems that **the data simply suggests that there is an influx of pro-inflammatory mediators from the plasma to the CSF following LPS injection.** This phenomenon is widely documented in the literature in the context of BBB leakiness. It is clear that the CPE can also produce exosomes, as was previously reported, and as the authors show in the *in vitro* model of figure 2, but the **relative contribution of this in comparison to BBB leakiness** was not addressed in the current manuscript, and is a major weakness of the suggested hypothesis. If the authors can address this question experimentally it would greatly improve the manuscript.

This question consists of three parts, which we have addressed in the revised manuscript and also discuss below: **(a)** the discrepancy between the observed EV increase in the CSF and gene expression analysis of exosome markers in choroid plexus, **(b)** the contribution of leakage of plasma components due to loss of blood-CSF barrier integrity, and **(c)** the importance of events that occur in parallel at the BBB.

(a) We observed a significant upregulation of EVs in the CSF already two hours after stimulation with LPS (i.p), while the mRNA upregulation of EV markers is detected only 8 h post LPS. This is not necessarily a discrepancy, as the choroid plexus epithelial cells of naive mice show multivesicular bodies (MVBs) with exosomes, which can be secreted without the need of new mRNA transcription. Moreover, at the first stages upon LPS injection, there might be enough protein available to increase the exosome production and secretion. Only later on, there will be the need to replenish the secreted EV proteins. To analyze EV protein levels in the choroid plexus epithelial cells, we performed **immunofluorescence analysis of different EV markers**, namely CD63, RAB5, and Annexin A2 (ANXA2), on brain sections of naive mice and 4 and 8 h after LPS

injection. As shown in **Figure 2**, this revealed a strong induction of the exosome proteins early upon stimulation with LPS. Interestingly, all three proteins were already expressed in basal conditions indicating that the necessary machinery is present to increase the EV production when needed.

Figure 2. Immunofluorescence analysis of CD63, RAB5 and ANXA2 in choroid plexus upon systemic inflammation. Representative confocal images of CD63, RAB5 and ANXA2 (red) at 0, 4 and 8 h after LPS treatment. Hoechst (blue) was used to stain the nucleus. The dotted line indicates the ependymal cells that line the ventricle and the square boxes indicate the zoomed insert images displayed at the right corner of each image. Scalebar 100 μ m. (CP, choroid plexus)

The immunostainings show that **CD63** is mainly observed in the perinuclear area in basal conditions and early upon LPS stimulation there is an increased signal at the apical side, close to the CSF. At a later time point, high CD63 levels are observed both at the perinuclear area and at the apical side of the choroid plexus epithelial cells. Similarly, **RAB5** is already present in the choroid plexus of naive mice and LPS stimulation results in higher levels of RAB5 both in the cytoplasm and at the apical side of the choroid plexus epithelial cells. Although **ANXA2** expression was less homogeneous throughout the choroid plexus, this marker is expressed at basal conditions and is strongly induced upon LPS stimulation. These results altogether suggest that the exosome machinery might be activated at the post-transcriptional level at the early time points after the inflammatory trigger. Only at later stages, to cope with the continuous secretion of the exosomes, there is the need to increase the mRNA levels. These results are included and discussed in the revised manuscript and replaced the qPCR data in Figure 4.

Moreover, we want to stress that we used different, independent experiments in our manuscript to prove that the choroid plexus epithelial cells are the main source of the EVs detected in the CSF:

(I) Transthyretin (TTR) is a protein that consists of four identical subunits of 14 kDa in a tetrahedral symmetry¹. Plasma TTR originates primarily in the liver, whereas brain TTR is exclusively produced, secreted, and regulated by the choroid plexus^{2,3}. We performed a western blot analysis on

EVs isolated from CSF and TTR was detected in all CSF-derived EV samples (**Figure 3**). This is in agreement with another study in which proteome analysis of CSF-derived EVs was optimized⁴.

Figure 3. Western blot analysis of choroid plexus cell lysate and extracellular vesicles isolated from CSF. Choroid plexus (CP) tissue was isolated, pooled from 3 mice, lysed, and analyzed by SDS-PAGE. Similarly, extracellular vesicles (EVs) were isolated from ~25 μ l CSF and analyzed by SDS-PAGE. Detection was done with an anti-TTR antibody (green) and an anti- β -actin antibody (red) using the Odyssey Imaging system.

(2) To prove the involvement of choroid plexus-derived exosomes in the observed LPS-dependent increase in the amount of CSF EVs, we made use of a chemical inhibitor, GW4869; i.e. a neutral sphingomyelinase inhibitor (nSMase2)⁵⁻⁸. Mice were injected intraperitoneally (ip) with LPS and 2.5 hours later injected intracerebroventricularly (icv) with GW4869 or vehicle. After 2.5 hours, CSF and choroid plexus were isolated and analyzed. CSF analysis by NanoSight revealed that inhibition of exosome production reduced the amount of EVs in the CSF (**Figure 4a**) and this was associated with a decrease in the secreted miRNAs miR-9, miR-146a, and miR-155 (**Figure 4b-d**). Moreover, this resulted in accumulation of several miRNAs in the choroid plexus: this accumulation was significant for several miRNAs (**Figure 4e-h**). These results show that blocking exosome production by icv injection of an exosome inhibitor prevents exosome release from the choroid plexus and leads to miRNA accumulation in the choroid plexus. This strongly supports that the choroid plexus is the main source of the observed LPS-dependent changes in EVs and miRNAs in the CSF.

Figure 4. Effect of exosome inhibition on EV and miRNA secretion of primary CPE cells stimulated with LPS. (a) In vitro quantification of EVs isolated from conditioned medium of LPS-stimulated primary CPE cells grown in a transwell system in the absence or presence of the exosome inhibitor GW4869 (n=3). (b-d) TaqMan assay quantification of the miRNAs miR-9 (b), miR-146a (c) and miR-155 (d) in supernatant of LPS-stimulated primary CPE cells grown in a transwell system and either left untreated or pretreated with GW4869 to inhibit exosome secretion (n=3). miR-1a levels were below detection limit. (e-h) TaqMan assay quantification of the

miRNAs miR-1a (e), miR-9 (f), miR-146a (g) and miR-155 (h) in cell lysate of LPS-stimulated primary CPE cells grown in a transwell system left untreated or treated with GW4869 to inhibit exosome secretion (n=3). Data are displayed as mean \pm SEM and analyzed by Student's t-test. Significance levels are indicated on the graphs: *, $0.01 \leq P < 0.05$; **, $0.001 \leq P < 0.01$.

(3) Additionally, we performed an *ex vivo* experiment by injecting mice with PBS or LPS and 2.5 hours later isolating choroid plexus after transcardial perfusion with PBS/heparin to remove all blood from the vascularized choroid plexus. Isolated choroid plexus explants were kept in culture for 2.5 hours in OptiMEM medium and supernatant was analyzed by NanoSight. This revealed the presence of significantly more EVs in the supernatant of choroid plexus from LPS-injected mice compared to PBS controls (Figure 5). Again, this provides evidence that the choroid plexus releases EVs into the CSF in response to peripheral LPS stimulation.

Figure 5. Analysis of choroid plexus explants from PBS- and LPS-injected mice. NanoSight analysis of supernatant of choroid plexus explants from PBS- or LPS-injected mice (n=6).

(4) We performed TEM analysis of choroid plexus tissue at different time points after LPS stimulation *in vivo* (Figures 6a–5i) and we quantified the amount of MVBs and exosomes per MVB and exosomes or intraluminal vesicles per cell. This provides evidence that choroid plexus epithelial cells are able to produce exosomes and that exosome production is increased upon systemic inflammation. Moreover, the CSF kinetics of the EVs (Figure 7a) resemble the kinetics in the choroid plexus epithelial cells (Figure 7b) quantified by TEM analysis and this further provides evidence that (part of) the EVs in the CSF are derived from the choroid plexus epithelial cells.

Figure 6. TEM analysis of choroid plexus from LPS-injected mice at different time points. (a-f) Representative TEM images from choroid plexus tissue isolated 0, 1, 2, 3, 4 or 6 hours after LPS injection. Quantification of (f) amount of MVBs per cell section, (g) amount of exosomes per MVB, and (h) amount of exosomes per cell section, based on TEM analysis.

Figure 7. Quantification of EVs in CSF and exosomes in the choroid plexus in response to peripheral LPS treatment. (a) NanoSight analysis of extracellular vesicles in the CSF 0, 1, 2, 4 and 6 h after LPS injection. **(b)** The amount of exosomes per cell section based on TEM analysis.

(5) Using the miRCURY LNA™ microRNA ISH Kit (Exiqon), we performed *in situ* hybridization (ISH) of three different miRNAs detected in EVs isolated from CSF. This revealed in all cases that the miRNAs were present in the choroid plexus cells in the cytoplasm (**Figure 8**).

Figure 8. In situ hybridization (ISH) analysis of miRNA expression in the choroid plexus. (a-c) LNA™-ISH of miR-146a (a), miR-9 (b) and miR-155 (c) on brain sections.

In conclusion, all data together support our hypothesis that the choroid plexus epithelial cells are the main source of the EVs that are detected in the CSF and all these data are in the manuscript.

(b) In the second part of the question, the reviewer mentions that LPS might induce an influx of pro-inflammatory mediators from the plasma into the CSF which might explain some of our observations. It is not completely clear what the reviewer means with ‘pro-inflammatory mediators’; we will focus here on the potential role of peripheral cytokines/chemokines and peripheral extracellular vesicles.

Previous data from our group indeed showed the presence of cytokines and chemokines in the CSF and disturbance of the blood-CSF barrier integrity upon systemic inflammation⁹. However, it is important to realize that these **cytokines/chemokines** are not a reflection of the peripheral levels arguing against direct leakage⁹, which suggests secretion by cells in the ventricles. Interestingly, LPS stimulation of primary choroid plexus epithelial cells cultured on transwells results in increased cytokine and chemokine levels in the upper compartment (**Figure 9**). Of course, the increased chemokine levels might subsequently induce leukocyte influx which might also contribute to the observed effects, but this will only occur at later time points upon LPS injection and is not responsible for the early increase in EVs in the CSF upon systemic inflammation. Importantly, although these cytokines and chemokines will play a role in the overall effects of systemic inflammation on the brain, this cytokine production is not the explanation of our findings. Indeed, when we isolate EVs, soluble proteins are not present in our isolated fraction.

Figure 9. Cytokine and chemokine analysis of supernatant of primary choroid plexus epithelial cells upon LPS stimulation. Primary choroid plexus cells were cultured on transwells and stimulated with LPS from the basal side. Supernatant from the upper compartment was isolated and TNF α , MCP1 and MIP1 α levels were determined using the Luminex technology (n=3).

Additionally, it is also important to realize that the reported increase in **blood-CSF barrier integrity** is determined using 4 kDa FITC-dextran⁹, while most pro-inflammatory mediators are larger. Next, we performed a new experiment with different molecular weight FITC-dextran molecules. Mice were injected i.p. with LPS, followed by i.v. injection of FITC-dextran. As presented in **Figure 10**, we observed increased blood-CSF barrier leakage of 4 kDa FITC-dextran, while this was not the case for 20 kDa FITC-dextran, suggesting that leakage of blood components will be limited.

Figure 10. Relative blood-CSF barrier leakage upon LPS injection. Mice were injected with PBS (n=3) or LPS (n=5), followed 3 h later by i.v. injection of 4 kDa and 20 kDa FITC-dextran. One hour later, CSF was isolated and fluorescence was measured using a fluorimeter ($\lambda_{ex}/\lambda_{em} = 488 \text{ nm}/520 \text{ nm}$).

Next to soluble blood components, also **peripheral extracellular vesicles** may cross the choroid plexus cells via transcytosis upon systemic inflammation, followed by release in the CSF, as shown for folate containing extracellular vesicles in naive mice¹⁰. We believe that this is a plausible (additional) mechanism, but this doesn't change anything to our data or hypothesis. Indeed, we show that choroid plexus-derived extracellular vesicles (either locally synthesized or transported across the choroid plexus epithelial cells from the blood via transcytosis) play an important role in the transfer of a pro-inflammatory message from blood to brain. Moreover, the fact that we can mimic the LPS-induced release of extracellular vesicles by primary choroid plexus cells *in vitro* together with the observation that the *in vivo* and *in vitro* EVs show a similar miRNA composition, suggests that most EVs in the CSF are newly synthesized by the choroid plexus epithelial cells. However, we do not exclude a role for transcytosis of peripheral EVs via the choroid plexus. We included this in the discussion of the revised manuscript.

(c) The last part of the question deals with the importance of **events that might happen in parallel at the blood-brain barrier (BBB)**. We do realize that the BBB is also important in the response to systemic LPS, and this is clearly stated in the discussion of the manuscript. For example, it is known that endothelial cell activation can lead to activation of other cells from the neurovascular unit that are closely connected to these endothelial cells. Therefore, we do not exclude the possibility that the endothelial cells can also produce EVs or that the endothelial response affects the choroid plexus in

response to peripheral inflammation. What we studied at the blood-CSF might occur in parallel with mechanisms that are activated at the BBB. However, we want to point out that morphometric studies of the BCSFB revealed that due to the presence of microvilli, choroid plexus epithelial cells have an apical surface area only twofold lower compared to the luminal surface area of the BBB endothelial cells, thereby providing an almost equally large surface for exchange of solutes and vesicles¹¹. Probably due to technical reasons, most researcher focus on the BBB and largely neglect the blood-CSF located at the choroid plexus. It is also important to keep in mind that the endothelial cells that form the BBB display minimal vesicle transport activity¹². In contrast, besides maintaining the barrier, the main function of the choroid plexus epithelial cells that form the blood-CSF barrier is secretion of proteins into the CSF. Indeed, choroid plexus epithelial cells at the blood-CSF display much more (vesicular) transport than endothelial BBB cells.

Additionally, the effects that we observe are really fast: as shown in **Figure 7a**, there was a gradual increase in the amount of EVs in the CSF and the increase is already significant 2 hours after LPS injection. Next, we analyzed the BBB integrity early upon LPS administration as described before¹³. We injected mice with LPS, followed by i.v. injection of 4 kDa FITC-dextran 3 hours later. Next, mice were perfused transcardially with PBS, brains were isolated and incubated in formamide to extract the fluid from the brain tissue. Analysis of the supernatant revealed that there was no increase in BBB leakage at this time point upon LPS injection (**Figure 11**), arguing against an important role of BBB leakage in the effects that we describe here. Moreover, the appearance of EVs in the CSF is unlikely to be a result of EVs that come from the endothelial cells, passed the brain parenchyma and entered the CSF in this short time frame. It is much more logic that the EVs originate from the choroid plexus epithelial cells which are in direct contact with the CSF and responsible for most CSF production. Additionally, as described above, we provide a lot of evidence that the choroid plexus epithelial cells are the main source of the EVs, based on the presence of TTR, TEM images, inhibitor studies, explant experiments, miRNA ISH and immunostainings of EV markers.

Figure 11. Relative BBB leakage upon LPS injection. Mice were injected with PBS (n=3) or LPS (n=6), followed 3 h later by i.v. injection of 4 kDa FITC-dextran. One hour later, brains were isolated and incubated in formamide overnight at 37° degrees. The next day, supernatant was isolated and fluorescence was measured using a fluorimeter ($\lambda_{ex}/\lambda_{em} = 488 \text{ nm}/520 \text{ nm}$).

The authors show data of several experiments in which they use the compound GW4869, which they refer to as "exosome inhibitor", to demonstrate that the pro-inflammatory effect and genes expression observed are exosome-dependent. Since GW4869 is a classical NFkB and TNF-alpha blocker, the observed effects are very much expected and do not contribute to the hypothesis presented. For example, in the data presented in figure 3, is it surprising that LPS-induced pro-

inflammatory genes expression is downregulated following exposure to an NFkB-blocker? This output repeats itself in various experiments along the manuscript, and should be cautiously regarded in terms of any mechanistic-insight to the exosomes.

GW4869 is a specific, non-competitive inhibitor of neutral sphingomyelinase (nSMase)¹⁴. Neutral sphingomyelinase helps in releasing the ceramide from the sphingomyelin and the secretion of exosomes from the cell. GW4869 is widely used in the literature as an exosome inhibitor, which is the reason why we have referred to GW4869 as an exosome inhibitor^{5, 6, 15-17}. We don't agree with the reviewer that the GW4869 is an NfκB blocker, as the literature claims the opposite effect: it has been shown that GW4869 doesn't affect TNF-induced NfκB translocation to nuclei and the TNF mediated signaling effects¹⁸.

In Figure 3 of our manuscript, we do not show any pro-inflammatory gene expression, so we are not sure what the reviewer is referring to. In these experiments, we used GW4869 on *in vitro* choroid plexus cells to see its effect on exosome production. This revealed that GW4869 suppresses the choroid plexus mediated exosome release into the supernatants upon LPS stimulation (Figure 3a in our manuscript). Additionally, this was associated with a decrease in exosome-associated miRNA release into the CSF (Figure 3b-d in our manuscript). Moreover, two out of four miRNAs were also increased in the choroid plexus epithelial cells upon GW4869 treatment (Figure 3e-h in our manuscript). These results show that blocking exosome production prevents exosome release from the choroid plexus and leads to miRNA accumulation in the choroid plexus.

We also used the exosome inhibitor in the experiments presented in Figure 4m-n of our original manuscript. Similar to our *in vitro* results presented in Figure 3, also *in vivo* icv injection inhibited exosome production and resulted into miRNA accumulation in the choroid plexus.

Finally, also Figure 8g-h of our manuscript contains some GW4869 data. As we observed a decrease in exosomes and exosome-associated miRNAs after treatment with GW4869 *in vitro* and *in vivo*, we wanted to analyze whether this eventually affects EV-mediated miRNA target -repression and pro-inflammatory gene expression in the brain. As shown, this resulted in increased expression of the miRNA targets and decreased levels of the pro-inflammatory genes, which shows that blockage of EV secretion reduces EV-mediated miRNA target repression and pro-inflammatory gene expression. Figure 6 shows massive amounts of data from protein analysis. Most of the data is not statistically significant (and is particularly accustomed to be presented in Z score and not P value, as per the multiple testing provided). Moreover, it is not appropriate to selectively choose from the GO enrichment terms specific biological processes of interest which fit a narrative (page 11), but to provide their significance. A short list of the top 5/10/X statistically significant changes would be much more informative to the reader.

We are a bit confused by this comment. We don't agree that most of the data is not significant, as shown by the p-values which are represented on the Y-axis. Only in case of Figure 6d, a limited amount of GO terms have a p-value between 0.1 and 0,05. This is expected, since this analysis is done on a limited set of proteins (namely 280; only the proteins which were detected upon LPS stimulation).

Additionally, we want to point out that we have not made any selection while doing the DAVID analysis; this is an unbiased DAVID GO enrichment analysis and we displayed the top pathways that came out of this analysis without selection. In Figure 6e, we grouped the top pathways in the three subgroups (namely 'biological processes', 'cellular components' and 'molecular functions') and in all three cases, the top 10 is displayed. Similarly, also the IPA analysis was done unbiased and the top pathways are displayed.

The data presented in figure 7 is important to the hypothesis presented in the manuscript, as it is basically the only link between the CSF exosomes, which have to go pass the ependymal layer, and the effect on the brain parenchyma. Unfortunately, the presented pictures are not convincing. Figure 7a,b show DAPI positive cells in the CSF, which are for some reason in proximity to GFAP positive astrocytes - where anatomically were these pictures taken? It is not clear at all from the picture how come the "CP", the "CSF", and the "brain" labeling are positioned where they are. There are specific immunohistochemistry markers which stain the ependymal layer of the brain ventricles; these should be used to show that the exosomes pass the ependymal layer. Also, to label the choroid plexus tissue, a specific marker should be used, such as TTR, or epithelial markers.

In our original images, we focused on regions in and around the ventricles to trace the injected EVs. The DAPI stained nuclei that the reviewer mentions in the comment are from the choroid plexus which is hanging inside the ventricle. However, in order to visually prove that these cells are indeed choroid plexus, and to improve the GFAP staining, we have repeated this experiment and performed a co-staining with pan-cytokeratin, an epithelial marker to visualize the choroid plexus. As shown in Figure 12b below, we found an abundant number of injected exosomes around the ventricle in the brain parenchyma, represented by the white arrowheads.

Further examination revealed that the PKH26 labeled exosomes (red) are very closely associated with the GFAP positive cells (**Figure 12, zoom**), indicating that the astrocytes surrounding the ventricles are actively involved in the uptake of the exosomes, similar to what was observed *in vitro*. We are convinced that the new images strengthen our hypothesis.

Figure 12. Tracing of intracerebroventricular (icv) injected EVs. Representative confocal images four hours after icv injection of PKH26 labelled EVs (red). Astrocytes are stained with GFAP (white) and nuclei with Hoechst (blue), and pan-cytokeratin (green) is used as the marker for the choroid plexus. The dotted line shows the ventricular border and the white arrow heads point to EVs that crossed the ependymal cell layer. Square box represents the zoomed image of a GFAP positive astrocyte. The experiment was repeated three times (n=3). Scalebar 100 μ m.

Minor comments:

In the legends to several figures the n values provided should be examined again. For example, figure 2c, 2e, and 3b, graphs have bars without standard errors, though in the legends it is described as n=3 or n=4. Were samples omitted from the groups? If so, please provide the specific n value for each group after samples were omitted. In several cases it looks like n=1 after samples were omitted. All sample size numbers are correctly provided in the legends of the revised manuscript. In all experiments, the n-value was ≥ 3 . If no error bar is visible, this is because the variation between the samples is small.

In the legends to the figures and the discussion there are several instances where *in vitro* experiments with LPS treatments are described as "systemic inflammation". It would be clearer if clearly stated as "LPS treatment".

We thank the reviewer for this suggestion and replaced 'systemic inflammation' by 'LPS treatment' in our revised manuscript.

Referee #3

Balusu et al. present a very interesting and original manuscript on the role of choroid plexus in the communication between the blood and the brain during peripheral inflammation, suggesting that the process is mediated by extracellular vesicles, mostly exosomes.

In addition to its originality and significance, the work shows a detailed and thorough description and characterisation of this phenomenon.

We thank the reviewer for these positive comments.

Some reservations relate to the recurring claim about the sensing capacity of the choroid plexus as no direct evidence is presented, only correlative data. At the very least, the authors should present evidence for the presence of receptors that can sense inflammation triggers, according to the model used. For example, the authors can try to use IHC to detect TLR4 in the case of LPS, or TNFR in the case of a TNF-driven model of inflammation. An alternative scenario is for example that either LPS or TNF alter the permeability of the BBB that then allows the same mediators to be sensed directly by microglia.

We appreciate the reviewer's comment. Indeed, the presence of TLR4 or TNFR1 in the choroid plexus is essential to assure a direct response to systemic inflammatory triggers such as LPS and TNF. First, we analyzed the expression of *Tlr4* and *Tnfr1* in the choroid plexus before and at different time points after LPS treatment and this revealed that both genes are expressed at the choroid plexus. As presented in **Figure 13**, we observed a significant induction of *Tnfr1* 6 and 24 h after LPS injection at mRNA level and *Tlr4* showed a limited increase in gene expression 6 h after LPS injection.

Figure 13. mRNA expression analysis of *Tlr4* and *Tnfr1* in the choroid plexus. Choroid plexus samples were isolated before and 1, 6 and 24 h post LPS treatment, followed by RNA isolation and mRNA expression analysis of *Tlr4* and *Tnfr1* (n=3).

Next, we performed immunostainings for both TNFR1 and TLR4. As shown in **Figure 14a**, TLR4 could be detected in the choroid plexus and showed an increase in the presence of LPS, in agreement with the qPCR results. TNFR1 signal was low in the choroid plexus of naive mice (**Figure 14b**), but increased upon LPS stimulation.

The reviewer also wonders whether LPS or TNF alter the BBB permeability which will activated the microglia. Although we do believe that, in parallel with what is happening at the blood-CSF barrier, also at the BBB several mechanisms are activated, we do know that the BBB is still intact 4 h after LPS injection. To address this, we analysed the BBB integrity upon LPS administration as described before¹³. We injected mice with LPS, followed by i.v. injection of 4 kDa FITC-dextran 3 hours later. Next, mice were perfused transcardially with DPBS, brains were isolated and incubated in formamide to extract the fluid from the brain tissue. Analysis of the supernatant revealed that there was no increase in BBB leakage at this time point upon LPS injection (**Figure 15**), arguing against a role for BBB leakage in the early stages of inflammation-induced EV secretion by the choroid plexus. However, we do not exclude that this might occur at later stages and might affect the choroid plexus.

Figure 15. Relative BBB leakage upon LPS injection. Mice were injected with PBS (n=3) or LPS (n=6), followed 3 h later by i.v. injection of 4 kDa FITC-dextran. One hour later, brains were isolated and incubated in formamide overnight at 37° degrees. The next day, supernatant was isolated and fluorescence was measured using a fluorimeter ($\lambda_{\text{ex}}/\lambda_{\text{em}} = 488 \text{ nm}/520 \text{ nm}$).

Another aspect that deserves clarification is the implication that up regulation of mir-146a contributes to inflammation as the literature largely favors the opposite effect.

We agree that anti-inflammatory properties have been attributed to miR-146a. While both miRNAs are coordinately regulated via similar mechanisms¹⁹, it has *e.g.* been shown that exosomal miR-146a inhibits, while miR-155 promotes endotoxin-induced inflammation in mice²⁰. As suggested by Alexander *et al.*, there are several possible reasons why EV populations contain both of these functionally distinct miRNAs species²⁰. One explanation is that miR-155 and miR-146a release in exosomes is a dynamically regulated process where the ratio of miR-155 to miR-146a changes over time and subsequently decides whether there will be a pro- or anti-inflammatory response in the target cells. However, further studies are needed to elucidate this. In our EVs, it is clear that miR-155 induction and expression is much higher compared to miR-146a, which might explain the pro-inflammatory effect on the recipient cells.

Finally, it would be interesting to investigate whether brain markers of inflammation are decreased in animals with genetic deficiencies in factors involved in exosome secretion such as Rab27.

We thank the reviewer for this enthusiastic question. Indeed, it would be really interesting to analyze the phenotype of Rab27 deficient mice in systemic inflammation. Unfortunately, we would need choroid plexus specific knockouts, since EV secretion in the periphery also plays an important role in systemic inflammation and these mice are not available. International Mouse Phenotyping Consortium has recently profiled the phenotype of full Rab27b knockout mice and found that these mice are severely defective in immune cell production. However, we do plan to study this in more detail in future projects by alternative approaches.

References

- Ingenbleek, Y. & Young, V. Transthyretin (prealbumin) in health and disease: nutritional implications. *Annu. Rev. Nutr.* **14**, 495-533 (1994).
- Aldred, A.R., Brack, C.M. & Schreiber, G. The cerebral expression of plasma protein genes in different species. *Comp. Biochem. Physiol. B Biochem. Mol. Biol.* **111**, 1-15 (1995).
- Herbert, J. et al. Transthyretin: a choroid plexus-specific transport protein in human brain. The 1986 S. Weir Mitchell award. *Neurology* **36**, 900-911 (1986).
- Chiasserini, D. et al. Proteomic analysis of cerebrospinal fluid extracellular vesicles: a comprehensive dataset. *J. Proteomics* **106**, 191-204 (2014).
- Li, J. et al. Exosomes mediate the cell-to-cell transmission of IFN-alpha-induced antiviral activity. *Nat. Immunol.* **14**, 793-803 (2013).

6. Kosaka, N. et al. Secretory mechanisms and intercellular transfer of microRNAs in living cells. *J. Biol. Chem.* **285**, 17442-17452 (2010).
7. Chairoungdua, A., Smith, D.L., Pochard, P., Hull, M. & Caplan, M.J. Exosome release of beta-catenin: a novel mechanism that antagonizes Wnt signaling. *J. Cell Biol.* **190**, 1079-1091 (2010).
8. Dinkins, M.B., Dasgupta, S., Wang, G., Zhu, G. & Bieberich, E. Exosome reduction in vivo is associated with lower amyloid plaque load in the 5XFAD mouse model of Alzheimer's disease. *Neurobiol. Aging* **35**, 1792-1800 (2014).
9. Vandenbroucke, R.E. et al. Matrix metalloprotease 8-dependent extracellular matrix cleavage at the blood-CSF barrier contributes to lethality during systemic inflammatory diseases. *J. Neurosci.* **32**, 9805-9816 (2012).
10. Grapp, M. et al. Choroid plexus transcytosis and exosome shuttling deliver folate into brain parenchyma. *Nature communications* **4**, 2123 (2013).
11. Keep, R.F. & Jones, H.C. A morphometric study on the development of the lateral ventricle choroid plexus, choroid plexus capillaries and ventricular ependyma in the rat. *Brain Res. Dev. Brain Res.* **56**, 47-53 (1990).
12. Stamatovic, S.M., Keep, R.F. & Andjelkovic, A.V. Brain endothelial cell-cell junctions: how to "open" the blood brain barrier. *Curr. Neuropharmacol.* **6**, 179-192 (2008).
13. Brkic, M. et al. Amyloid beta Oligomers Disrupt Blood-CSF Barrier Integrity by Activating Matrix Metalloproteinases. *J. Neurosci.* **35**, 12766-12778 (2015).
14. Lee, D.H. et al. Identification and evaluation of neutral sphingomyelinase 2 inhibitors. *Arch. Pharm. Res.* **34**, 229-236 (2011).
15. Ostrowski, M. et al. Rab27a and Rab27b control different steps of the exosome secretion pathway. *Nat. Cell Biol.* **12**, 19-30; sup pp 11-13 (2010).
16. Essandoh, K. et al. Blockade of exosome generation with GW4869 dampens the sepsis-induced inflammation and cardiac dysfunction. *Biochim. Biophys. Acta* **1852**, 2362-2371 (2015).
17. Yuyama, K., Sun, H., Mitsutake, S. & Igarashi, Y. Sphingolipid-modulated exosome secretion promotes clearance of amyloid-beta by microglia. *J. Biol. Chem.* **287**, 10977-10989 (2012).
18. Luberto, C. et al. Inhibition of tumor necrosis factor-induced cell death in MCF7 by a novel inhibitor of neutral sphingomyelinase. *J. Biol. Chem.* **277**, 41128-41139 (2002).
19. Doxaki, C., Kampranis, S.C., Eliopoulos, A.G., Spilianakis, C. & Tsatsanis, C. Coordinated Regulation of miR-155 and miR-146a Genes during Induction of Endotoxin Tolerance in Macrophages. *J. Immunol.* **195**, 5750-5761 (2015).
20. Alexander, M. et al. Exosome-delivered microRNAs modulate the inflammatory response to endotoxin. *Nature communications* **6**, 7321 (2015).

2nd Editorial Decision

15 June 2016

Thank you for the submission of your revised manuscript to EMBO Molecular Medicine. We have now received the enclosed reports from the referees that were asked to re-assess it. As you will see the reviewers are now globally supportive and I am pleased to inform you that we will be able to accept your manuscript pending the following final amendments:

1) Please address the minor change commented by referee 1. Please provide a letter INCLUDING the reviewer's reports and your detailed responses to their comments (as Word file).

Please submit your revised manuscript within two weeks. I look forward to seeing a revised form of your manuscript as soon as possible.

***** Reviewer's comments *****

Referee #2 (Remarks):

The revised manuscript is considerably improved, in particularly by the newly added data in figures

4, 7, and supplementary EV5 and EV12, which strengthen the authors hypothesis for the CP as a significant source of the exosomes. These, and the question regarding the contribution of BBB leakage, were major limitations of the original manuscript, which the authors adequately and thoroughly addressed.

With regards to figure 6, which the authors did not revise - I believe it still doesn't say much to the reader, except that perhaps "many things are changing in response to LPS". Preferably, focusing on the top important or statistically significant data out of the David software output, would make it clearer.

Referee #3 (Remarks):

The authors have satisfactorily addressed all my concerns. This is now a much important and stronger manuscript!

2nd Revision - authors' response

11 July 2016

1. Please address the minor change commented by referee 1. Please provide a letter INCLUDING the reviewer's reports and your detailed responses to their comments (as Word file).

We assume this is about the following comment of referee #2: "With regards to figure 6, which the authors did not revise - I believe it still doesn't say much to the reader, except that perhaps "many things are changing in response to LPS". Preferably, focusing on the top important or statistically significant data out of the David software output, would make it clearer." To address this comment, we removed the non-significant data from the graphs in Figure 6.

Corresponding Author Name: Roosmarijn Vandenbroucke

Manuscript Number: EMM-2016-06271